# Kantorovich Strikes Back!
# Wasserstein GANs are not Optimal Transport?

**Alexander Korotin**[*]
Skolkovo Institute of Science and Technology
Artificial Intelligence Research Institute
Moscow, Russia
`a.korotin@skoltech.ru`

**Alexander Kolesov\***
Skolkovo Institute of Science and Technology
Moscow, Russia
`a.kolesov@skoltech.ru`

**Evgeny Burnaev**
Skolkovo Institute of Science and Technology
Artificial Intelligence Research Institute
Moscow, Russia
`e.burnaev@skoltech.ru`

## Abstract

Wasserstein Generative Adversarial Networks (WGANs) are the popular generative models built on the theory of Optimal Transport (OT) and the Kantorovich duality. Despite the success of WGANs, it is still unclear how well the underlying OT dual solvers approximate the OT cost (Wasserstein-1 distance, $\mathbb{W}_1$) and the OT gradient needed to update the generator. In this paper, we address these questions. We construct 1-Lipschitz functions and use them to build ray monotone transport plans. This strategy yields pairs of continuous benchmark distributions with the analytically known OT plan, OT cost and OT gradient in high-dimensional spaces such as spaces of images. We thoroughly evaluate popular WGAN dual form solvers (gradient penalty, spectral normalization, entropic regularization, etc.) using these benchmark pairs. Even though these solvers perform well in WGANs, none of them faithfully compute $\mathbb{W}_1$ in high dimensions. Nevertheless, many provide a meaningful approximation of the OT gradient. These observations suggest that these solvers should not be treated as good estimators of $\mathbb{W}_1$, but to some extent they indeed can be used in variational problems requiring the minimization of $\mathbb{W}_1$.

The Wasserstein-1 distance [3] ($\mathbb{W}_1$) is a popular loss function to learn generative models. It has numerous advantages compared to the vanilla GAN loss [15]. For example, $\mathbb{W}_1$ is correctly defined if the distributions' supports differ [2]. Besides, it correlates with the sample quality, provides improved stability of the optimization process and does not suffer from the vanishing gradients issue [3, §4].

Generative models which employ $\mathbb{W}_1$ as the loss to update the generator are called the Wasserstein GANs (WGANs). To compute $\mathbb{W}_1$, they use its variational approximation based on the **Kantorovich duality** [20] and the **Optimal Transport** (OT) theory [52, 47]. Since the introduction of the original WGANs with the weight clipping method [3], a lot of alternative techniques (*neural dual OT solvers*) to compute $\mathbb{W}_1$ have been proposed: gradient penalties [17, 39, 54], entropic regularization [46], architectural constraints [35, 1], batch-based methods [31, 29], maximin methods [38, 23], etc.

Despite the popularity of WGANs, it still remains unclear to what extent their success is connected to OT and $\mathbb{W}_1$ rather than, e.g., to a good choice of regularization [49, §7]. Due to the limited amount of pairs of distributions with known $\mathbb{W}_1$, it is challenging to evaluate existing dual OT solvers.

**Contributions.** We develop a generic methodology based on the *transport rays* (§3.1) to evaluate dual OT solvers for the Wasserstein-1 distance ($\mathbb{W}_1$). Our main contributions are as follows:

---

[*]Equal contribution.

36th Conference on Neural Information Processing Systems (NeurIPS 2022) Track on Datasets and Benchmarks.

- We use 1-Lipschitz functions to construct pairs of *continuous* distributions that we use as a benchmark with analytically-known OT cost, map and gradient for $\mathbb{W}_1$ transport (§3.2, §3.3).

- We use these *benchmark distributions* to evaluate (§4) popular WGAN dual OT solvers (§2) in high-dimensional spaces, including the spaces of $32 \times 32$ CIFAR-10 images, $64 \times 64$ CelebA faces.

Related works [32, 49] consider *discrete* distributions and show that some solvers fail to estimate $\mathbb{W}_1$. In contrast to them, we study how well the solvers compute the gradient of $\mathbb{W}_1$ (*OT gradient*), as it is the OT gradient which is used to update the generator in WGANs, not the value of $\mathbb{W}_1$. We use *continuous* distributions since in the discrete case the OT gradient may be ill-defined (§1).

**Notation.** We work in the $\mathbb{R}^D$ space that is endowed with the Euclidean norm $||\cdot||_2$. We use $\mu_L$ to denote the Lesbegue measure on $\mathbb{R}^D$. For a measurable map $T : \mathbb{R}^D \to \mathbb{R}^D$, we denote the associated pushforward operator by $T\sharp$. We consider Borel probability distributions $\mathbb{P}, \mathbb{Q}$ on $\mathbb{R}^D$ with finite first moments. We use $\Pi(\mathbb{P}, \mathbb{Q})$ to denote the set of probability distributions on $\mathbb{R}^D \times \mathbb{R}^D$ with marginals $\mathbb{P}$ and $\mathbb{Q}$ (transport plans). All the integrals are computed over $\mathbb{R}^D$, if not stated otherwise. We write $||f||_L \leq C$ if $f : \mathbb{R}^D \to \mathbb{R}$ is $C$-Lipschitz.

# 1 Background on Optimal Transport

**Primal Formulation.** For distributions $\mathbb{P}, \mathbb{Q}$, the **Monge**'s formulation of the Wasserstein-1 ($\mathbb{W}_1$) distance, i.e., OT with the distance cost function $||x - y||_2$, is given by (Figure 1a)

$$\mathbb{W}_1(\mathbb{P}, \mathbb{Q}) \overset{def}{=} \min_{T\sharp\mathbb{P}=\mathbb{Q}} \int ||x - T(x)||_2 d\mathbb{P}(x), \tag{1}$$

where min is taken over measurable functions $T : \mathbb{R}^D \to \mathbb{R}^D$ (transport maps) that map $\mathbb{P}$ to $\mathbb{Q}$. The optimal $T^*$ is called the *optimal transport map* (OT map). Note that (1) is not symmetric, and this formulation does not allow for mass splitting, i.e., for some $\mathbb{P}, \mathbb{Q}$, there is no map $T$ that satisfies $T\sharp\mathbb{P} = \mathbb{Q}$ [40, Remark 2.4]. Thus, **Kantorovich** [20] proposed the following relaxation (Figure 1b):

$$\mathbb{W}_1(\mathbb{P}, \mathbb{Q}) \overset{def}{=} \min_{\pi \in \Pi(\mathbb{P}, \mathbb{Q})} \int_{\mathbb{R}^D \times \mathbb{R}^D} ||x - y||_2 d\pi(x, y), \tag{2}$$

where min is taken over transport plans $\pi \in \Pi(\mathbb{P}, \mathbb{Q})$. The optimal $\pi^* \in \Pi(\mathbb{P}, \mathbb{Q})$ is called the *optimal transport plan* (OT plan). If $\pi^* = [\mathrm{id}_{\mathbb{R}^D}, T^*]\sharp\mathbb{P}$ for some map $T^* : \mathbb{R}^D \to \mathbb{R}^D$, then $T^*$ minimizes formulation (1). In general, there might exist more than one OT plan $\pi^*$ or OT map $T^*$.

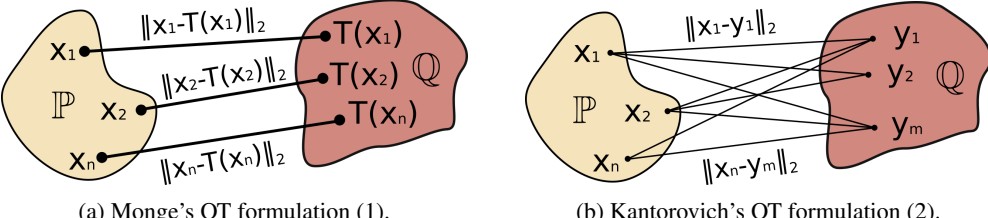

(a) Monge's OT formulation (1).  (b) Kantorovich's OT formulation (2).

Figure 1: Monge's and Kantorovich's OT fomulations of the Wasserstein-1 distance ($\mathbb{W}_1$).

**Dual formulation.** For distributions $\mathbb{P}, \mathbb{Q}$, the dual formulation of $\mathbb{W}_1$ is given by [52, Thm. 5.10]:

$$\mathbb{W}_1(\mathbb{P}, \mathbb{Q}) = \max_{f \oplus g \leq ||\cdot||_2} \int f(x) d\mathbb{P}(x) + \int g(y) d\mathbb{Q}(y), \tag{3}$$

where max is taken over $f, g : \mathbb{R}^D \to \mathbb{R}$ satisfying $f(x) + g(y) \leq ||x - y||_2$ for all $x, y \in \mathbb{R}^D$. By using the $c$-transform $f^c(y) \overset{def}{=} \min_{x \in \mathbb{R}^D} \{||x - y||_2 - f(x)\}$ [52, §5], one rewrites (3) as

$$\mathbb{W}_1(\mathbb{P}, \mathbb{Q}) = \max_f \int f(x) d\mathbb{P}(x) + \int f^c(y) d\mathbb{Q}(y). \tag{4}$$

In accordance with [52, Case 5.16], dual form (4) can be further restricted to 1-Lipschitz functions. In this case, it holds $f^c(y) = -f(y)$ [52, Case 5.4], and the alternative duality formula for $\mathbb{W}_1$ is

$$\mathbb{W}_1(\mathbb{P}, \mathbb{Q}) = \max_{||f||_L \leq 1} \int f(x) d\mathbb{P}(x) - \int f(y) d\mathbb{Q}(y). \tag{5}$$

In WGAN literature [17, 2], function $f$ is typically called the *critic* (or discriminator). In OT literature [52, 47, 51], functions $f, f^c, g$ are commonly refered as the (Kantorovich) *potentials*.

**Optimal transport in GANs.** Derivatives of $\mathbb{W}_1$ are used implicitly in generative modeling [3, 39, 35, 17, 38, 46, 48, 28] that incorporates $\mathbb{W}_1$ loss, in which case $\mathbb{P} = \mathbb{P}_\alpha$ is a parametric distribution and $\mathbb{Q}$ is the data distribution. Typically, $\mathbb{P}_\alpha = G_\alpha \sharp \mathbb{S}$ is the distribution generated from a fixed latent distribution $\mathbb{S}$ by a generator network $G_\alpha$. The goal is to find parameters $\alpha$ that minimize $\mathbb{W}_1(\mathbb{P}_\alpha, \mathbb{Q})$ via gradient descent. The loss function for the generator is:

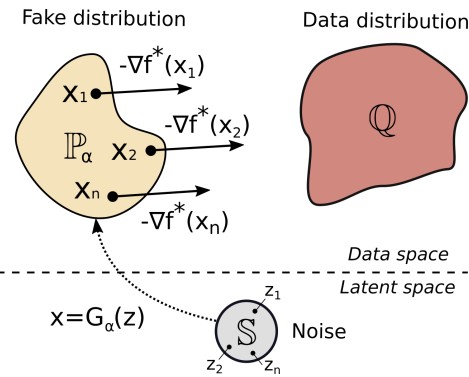

$$\mathbb{W}_1(\mathbb{P}_\alpha, \mathbb{Q}) = \int_z f^*(G_\alpha(z)) d\mathbb{S}(z) - \int f^*(y) d\mathbb{Q}(y),$$

where $f^*$ is the optimal potential in (5). The loss derivative (Figure 2) is given by [13, Eq. 3]:

$$\frac{\partial \mathbb{W}_1(\mathbb{P}_\alpha, \mathbb{Q})}{\partial \alpha} = \int_z \mathbf{J}_\alpha G_\alpha(z)^T \nabla f^*(G_\alpha(z)) d\mathbb{S}(z),$$

Figure 2: The anti-gradient $-\nabla f^*(x)$ shows where to move the mass of each $x = G_\alpha(z)$ to make the generated $\mathbb{P}_\alpha$ closer to $\mathbb{Q}$ in $\mathbb{W}_1$.

where $\mathbf{J}_\alpha G_\alpha(z)^T$ is the transpose of the Jacobian matrix of $G_\alpha(z)$ w.r.t. parameters $\alpha$. This result still holds without assuming the potentials are fixed [3, Theorem 3] by the envelope theorem [33].

In practice, the optimal potential $f^*$ is unknown. Therefore, WGANs approximate it with a network $f_\theta : \mathbb{R}^D \to \mathbb{R}$ (potential) by maximizing (5), (4) or (3) via the stochastic gradient ascent (SGA). This is usually associated with *evaluating* the $\mathbb{W}_1$ loss. However, note that *the loss value plays no role* in generator updates. Only the gradient $\nabla f^*$ of the potential is needed. We call it the *OT gradient*. We use a generic phrase *OT solver* to refer to any algorithm which is capable of recovering $\nabla f^*$.

**Quantitative evaluation of OT solvers.** Existing solvers are typically tested as the loss in WGANs without evaluating the actual OT performance. The quality of the generated samples is evaluated by standard metrics such as FID [18] or IS [45]. These metrics do not provide understanding about the quality of the solver itself since they depend on components of the model that are not related to OT.

In [41, 32, 49], the authors use discrete $\mathbb{P}, \mathbb{Q}$ to show that some solvers imprecisely compute $\mathbb{W}_1$. Their approach is *not applicable* to evaluation of the OT gradient, as $\nabla f^*$ is ill-defined in the discrete case. For example, when $\mathbb{P} = \delta_0$, $\mathbb{Q} = \delta_1$, it holds that $f^* = -[x]_+$ is an optimal potential, but it is not even differentiable at $x = 0 = \text{Supp}(\mathbb{P})$. The existence of the OT gradient is studied, e.g., in [19].

## 2 Neural Dual Solvers for the Wasserstein-1 Distance

Our proposed benchmark is useful for testing any OT solver that computes $\nabla f^*$ or $\mathbb{W}_1$. We evaluate only neural solvers which are based on (5), (4), or (3) and used in WGANs. We provide an overview of these methods below. We group them by the dual formulations which they use.

Most solvers approximate the potential by a network $f_\theta : \mathbb{R}^D \to \mathbb{R}$ and learn it via maximizing (5) with SGA on batches from $\mathbb{P}, \mathbb{Q}$. The main challenge is to enforce the 1-Lipschitz constraint for $f_\theta$.

$\lfloor$**WC**$\rfloor$ In [3], the space $\Theta$ of parameters is restricted to a compact, e.g., to a hypercube $[-c, c]^{\dim \theta}$. With mild assumptions on the architecture, $f_\theta$ is provably Lipschitz continuous with some *unknown* constant $C$, i.e., $||f_\theta||_L \leq C$. The main practical issue is tuning the boundary $c$ of the set.

$\lfloor$**GP**$\rfloor$ The authors of [17] prove that with mild assumptions on the OT plan $\pi^*$, the equation $||\nabla f^*(z)||_2 = 1$ holds almost surely for $z = tx + (1-t)y$ with $(x, y)$ distributed as the OT plan $\pi^*$ and $t \sim \text{Uniform}[0, 1]$. Thus, they *softly* penalize $f$ for being not 1-Lipschitz and optimize

$$\mathbb{W}_1(\mathbb{P}, \mathbb{Q}) \approx \max_f \left\{ \int f(x) d\mathbb{P}(x) - \int f(y) d\mathbb{Q}(y) - \lambda \mathcal{R}(f) \right\}, \quad \lambda > 0. \qquad (6)$$

The *gradient penalty* $\mathcal{R}_{\text{GP}}(f)$ equals $\int (||\nabla f(r)||_2 - 1)^2 d\mu(r)$, where $\mu$ is the distribution of a random variable $r = xt + (1-t)y$ with $t \sim \text{Uniform}[0, 1]$ and $(x, y) \sim \mathbb{P} \times \mathbb{Q}$ (as $\pi^*$ is unknown). In [34], it is proved that (6) with $\mathcal{R} = \mathcal{R}_{\text{GP}}$ is a specific OT formulation called *congested* OT.

$\lfloor$**LP**$\rfloor$ In [39], the authors show that $\mathcal{R}_{\text{GP}}(f)$ suffers from instabilities and high magnitudes. They introduce the *Lipschitz penalty* $\mathcal{R}_{\text{LP}}(f) = \int (\max\{0, ||\nabla f(r)||_2 - 1\})^2 d\mu(r)$ resolving the issues.

$\lfloor$**SN**$\rfloor$ In [35], the authors optimize (5) and use the power iteration method [14] to normalize the weight matrices of linear layers of net $f_\theta$ by their spectral norms. This provably makes $f$ globally Lipschitz continuous, which does not hold for [17, 39] which are based on the soft penalization (6).

$\lfloor$**SO**$\rfloor$ The authors of [1] claim that spectral normalization negatively affects the expressiveness of the network. They propose to ortho-normalize weight matrices and use GroupSort activations [5]. Such networks $f_\theta$ are 1-Lipschitz and universally approximate 1-Lipschitz functions [1, §4].

Below we overview methods based on (4) or (3) which do not require enforcing 1-Lipschitz continuity for the potential. Unlike the above-mentioned methods, they mostly require 2 neural networks.

$\lfloor$**LS**$\rfloor$ In [46, 48, 12, 6], the authors optimize the following *unconstrained* regularized form (3):

$$\mathbb{W}_1(\mathbb{P}, \mathbb{Q}) \approx \max_{f,g} \left\{ \int f(x) d\mathbb{P}(x) + \int g(y) d\mathbb{Q}(y) - \mathcal{R}(f, g) \right\}, \tag{7}$$

where $\mathcal{R}(f, g)$ is the *entropic* or *quadratic* regularizer [46, Eq.5] which softly penalizes the potentials $f, g$ for disobeying $f \oplus g \le || \cdot ||_2$. In practice, $f, g$ are neural networks $f_\theta, g_\omega : \mathbb{R}^D \to \mathbb{R}$.

$\lfloor$**MM:B**$\rfloor$ The authors of [31, 32] expand the dual formulation (4) with the $c$-transform:

$$\mathbb{W}_1(\mathbb{P}, \mathbb{Q}) = \max_f \left\{ \int f(x) d\mathbb{P}(x) + \int \min_{x \in \mathbb{R}^D} [\|x - y\|_2 - f(y)] d\mathbb{Q}(y) \right\}. \tag{8}$$

During optimization, they restrict the inner minimization to the current mini-batch from $\mathbb{P}$. This leads to *overestimation* of the inner problem's solution since the minimum is taken over a restricted subset. Recently, a more tricky version of this approach appeared [27, §3]. We call it $\lfloor$**MM:Bv2**$\rfloor$.

$\lfloor$**MM**$\rfloor$ In [38], the authors use a saddle point formulation equivalent to (8):

$$\mathbb{W}_1(\mathbb{P}, \mathbb{Q}) = \max_f \int f(x) d\mathbb{P}(x) + \min_H \int \left[ \|H(y) - y\|_2 - f(H(y)) \right] d\mathbb{Q}(y), \tag{9}$$

where the minimization is performed over functions $H : \mathbb{R}^D \to \mathbb{R}^D$. The authors use neural networks $f_\theta$ and $H_\omega$ to parametrize the potential and the minimizer of the inner problem (*mover*). To train $\theta, \omega$, the authors apply stochastic gradient ascent/descent (SGAD) over mini-batches from $\mathbb{P}, \mathbb{Q}$.

$\lfloor$**MM:R**$\rfloor$ One may also recover the OT gradient $\nabla f^*$ from the OT map $T^*$. Consider the form

$$\mathbb{W}_1(\mathbb{P}, \mathbb{Q}) = \max_g \min_T \int \left[ \|T(x) - x\|_2 - g(T(x)) \right] d\mathbb{P}(x) + \int g(y) d\mathbb{Q}(y), \tag{10}$$

which is a *reversed* [23] version of (9), i.e., the roles of $\mathbb{P}, \mathbb{Q}$ are swapped and $T : \mathbb{R}^D \to \mathbb{R}^D$. For *some* optimal saddle points $(g^*, T^*)$ of (10) it holds that mover $T^*$ is an OT map [25, Lemma 4], [11, Lemma 2], [50], [8]. With mild assumptions on $\mathbb{P}, \mathbb{Q}$, one may recover $T^*$ and use the identity $\nabla f^*(x) = \frac{x - T^*(x)}{\|x - T^*(x)\|_2}$ [7, §1] to obtain the OT gradient from $T^*$.

## 3 Constructing Benchmark Distributions for Dual Solvers

In this section, we develop a generic methodology to construct pairs $(\mathbb{P}, \mathbb{Q})$ with computable ground truth OT plan, OT cost and OT gradient. Our approach is inspired by the insights about OT plans in [47, §3.1], [16], [7, §3-7]. In §3.1, we give the required preliminaries. In §3.2, we provide our method to build benchmark pairs. We construct them in §3.3. The proofs are given in Appendix A.

### 3.1 Ray Monotone Transport Plans and 1-Lipschitz Functions

Let $u : \mathbb{R}^D \to \mathbb{R}$ be a 1-Lipschitz function. Recall that due to the Rademacher's theorem, $u$ is differentiable $\mu_L$-almost everywhere. For every $x, y \in \mathbb{R}^D$ satisfying $u(x) - u(y) = \|x - y\|_2$ it holds that $u$ is affine on the segment $[x, y]$, i.e., $u(z) = tu(x) + (1 - t)u(y)$ for $z = tx + (1 - t)y$ with $t \in [0, 1]$ [47, Lemma 3.5]. Moreover, $u$ is differentiable at all $z \in (x, y)$ and $\nabla u(z) = \frac{x - y}{\|x - y\|_2}$ [47, Lemma 3.6]. Following [47, Definition 3.7], we call a *transport ray* any non-trivial (different from a singleton) segment $[x, y]$ such that $u(x) - u(y) = \|x - y\|_2$, which is maximal for the inclusion among segments of this form. The unit vector $\frac{x - y}{\|x - y\|_2}$ is called the direction of a transport ray. Two transport rays can only intersect at their boundary points [46, Corollary 3.8]. In general, not all points

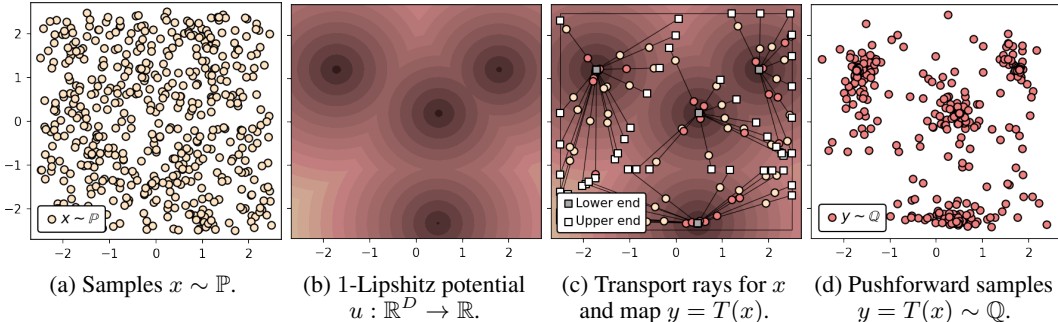

(a) Samples $x \sim \mathbb{P}$.

(b) 1-Lipshitz potential $u : \mathbb{R}^D \to \mathbb{R}$.

(c) Transport rays for $x$ and map $y = T(x)$.

(d) Pushforward samples $y = T(x) \sim \mathbb{Q}$.

Figure 3: Our methodology to construct benchmark pairs $(\mathbb{P}, \mathbb{Q})$. An example with $(D, N) = (2, 4)$. We pick a distribution $\mathbb{P}$ and a MinFunnel function $u : \mathbb{R}^D \to \mathbb{R}$ (11). To sample $y \sim \mathbb{Q}$, we get $x \sim \mathbb{P}$, compute its transport ray and move $x$ along the ray with the $u$-ray monotone map $T$ (12).

belong to transport rays. For example, for the function $u(x) = \frac{1}{2}x$ there are no transport rays at all. In Figures 3c and 9, we provide examples of transport rays for 1-Lipschitz functions $u$ in $D = 2$.

For a 1-Lipschitz function $u : \mathbb{R}^D \to \mathbb{R}$, we say that a transport plan $\pi \in \Pi(\mathbb{P}, \mathbb{Q})$ is $u$-*ray monotone* (decreasing) if $u(x) - u(y) = \|x - y\|_2$ holds $\pi$-almost surely for $x, y \in \mathbb{R}^D$. The idea of the definition is that such plans distribute the probability mass of $x \sim \mathbb{P}$ among $y \in \mathbb{R}^D$ such that $y = x$ (no mass movement) and/or $y$ which lie on the same transport ray as $x$ but below $x$, i.e., $u(y) < u(x)$. If $x$ is not contained in a transport ray, then $\pi(y|x) = \delta_x$, i.e., $\pi$ necessarily does not move $x$.

**Proposition 1** (Ray monotone transport plans are optimal). *Let $\pi \in \Pi(\mathbb{P}, \mathbb{Q})$ be a $u$-ray monotone transport plan for a* 1-*Lipschitz function $u$. Then it is an optimal plan between $\mathbb{P}, \mathbb{Q}$. Besides, $u$ is an optimal potential, i.e., it attains the maximum in dual formulation* (5).

For a distribution $\mathbb{P}$, we say that distribution $\mathbb{Q}$ is a $u$-*ray-forward* of $\mathbb{P}$ if there exists a measurable function $T : \mathbb{R}^D \to \mathbb{R}^D$ satisfying $T\sharp\mathbb{P} = \mathbb{Q}$ and $u(x) - u\big(T(x)\big) = \|x - T(x)\|_2$ holds $\mathbb{P}$-almost surely for all $x \in \mathbb{R}^D$. We say that such a $T$ is a $u$-*ray-monotone* transport map from $\mathbb{P}$ to $\mathbb{Q}$. Note that the deterministic plan $\pi = [\mathrm{id}_{\mathbb{R}^D}, T]\sharp\mathbb{P}$ is $u$-ray monotone. We have the following corollary:

**Corollary 1** (Ray monotone transport maps are optimal). *Let $T$ be a $u$-ray monotone transport map from $\mathbb{P}$ to $\mathbb{Q}$. Then the plan $\pi = [\mathrm{id}_{\mathbb{R}^D}, T]\sharp\mathbb{P}$ is optimal and $T$ is an OT map from $\mathbb{P}$ to $\mathbb{Q}$.*

In §3.2 below, we derive our recipe to construct **benchmark pairs** $(\mathbb{P}, \mathbb{Q})$ such that $\mathbb{Q}$ is a $u$-ray-forward of $\mathbb{P}$ with user-defined $u$ and analytically known $T$. In such pairs $\mathbb{P}$ is accessible by samples and is also possible to sample from $\mathbb{Q}$ by pushing $x \sim \mathbb{P}$ forward by $T$. Since $T$ is an OT map, the ground truth OT cost is $\mathbb{W}_1(\mathbb{P}, \mathbb{Q}) = \int \|x - T(x)\|_2 d\mathbb{P}(x)$. It admits *unbiased* Monte Carlo estimates from samples $x \sim \mathbb{P}$. Moreover, with mild assumptions (§3.3), $\nabla u$ is the *unique* OT gradient. Our benchmark pairs can be used to test how well OT solvers recover $\mathbb{W}_1$ and the OT gradient.

## 3.2 Method to Construct Benchmark Pairs

Let $u$ be a known 1-Lipschitz function. We aim to find its transport rays and construct a $u$-ray-forward map $T$ and distribution $\mathbb{Q} = T\sharp\mathbb{P}$ (for a given $\mathbb{P}$) for testing dual OT solvers. First, we describe the parametric class of 1-Lipschitz functions $u$ which we employ in our benchmark. Second, we explain how to compute the transport rays of $u$. Finally, we explain how to define $u$-ray monotone maps.

**Part 1. Parameterizing** 1-**Lipschitz functions.** Inspired by the distance representation of optimal potentials [16], as $u : \mathbb{R}^D \to \mathbb{R}$, we employ the following 1-Lipschitz *MinFunnel* functions:

$$u(x) \overset{def}{=} \min_n \{u_n(x)\} = \min_n \{\|x - a_n\|_2 + b_n\}, \tag{11}$$

where $a_n \in \mathbb{R}^D$ and $b_n \in \mathbb{R}$ are the parameters. Each *funnel* $u_n(x) = \|x - a_n\|_2 + b_n$ has $\|\nabla u_n(x)\|_2 = 1$ when $x \neq a_n$. Thus, $u$ is also 1-Lipschitz as it is their minimum. Note that for $\mu_L$-almost every $x$ it holds that $u$ is differentiable at $x$ and $\|\nabla u(x)\|_2 = 1$.

**Proposition 2** (MinFunnels are universal approximators of 1-Lipschitz functions on compact sets). *Let $\mathcal{S} \subset \mathbb{R}^D$ be a compact set and $f^* : \mathcal{S} \to \mathbb{R}$ be a 1-Lipschitz function. Then for every $\epsilon > 0$ there exists $N$ and $\{a_n, b_n\}_{n=1}^N$ such that function* (11) *satisfies* $\sup_{x \in \mathcal{S}} |u(x) - f^*(x)| \leq \epsilon$.

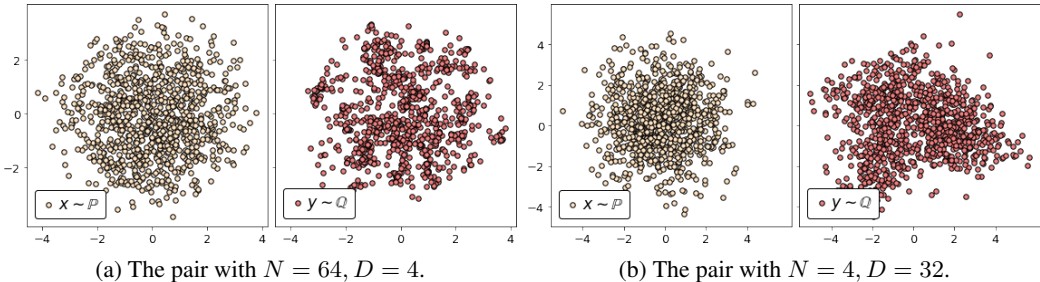

(a) The pair with $N = 64, D = 4$.  (b) The pair with $N = 4, D = 32$.

Figure 4: Visualization of constructed high-dimensional benchmark pairs.
In each pair, we show random samples projected onto 2 principal components of $\mathbb{Q}$.

The proposition is included only for completeness of the exposition and is not principal for our construction. We do not aim to approximate a specific $f^*$ by $u$. In practice, we found that simply picking a *random* MinFunnel $u$ and using it to construct a benchmark pair is reasonable (§3.3).

Part 2. **Computing the transport rays.** MinFunnels are practically very convenient as the transport ray for a given $x \in \mathbb{R}^D$ can be analytically computed in $O(ND)$ time, see our proposition below.

**Proposition 3** (Transport rays of a MinFunnel). *Consider MinFunnel* (11) *function* $u : \mathbb{R}^D \to \mathbb{R}$ *with $n$ distinct centers $a_n$ such that for all $n_1 \neq n_2$ it holds that $\|a_{n_1} - a_{n_2}\|_2 \neq |b_{n_1} - b_{n_2}|$. Let $x \in \mathbb{R}^D$ be a point such that $u$ is differentiable at $x$. Then $ray(x) = [a_m, x + r \cdot v]$, where*

$$m \stackrel{def}{=} \arg\min_n \{u_n(x)\}; \qquad v \stackrel{def}{=} \frac{x - a_m}{\|x - a_m\|_2}; \qquad r \stackrel{def}{=} \min_n r_n \in \mathbb{R} \cup \{+\infty\},$$

*and* $r_n \stackrel{def}{=} \frac{1}{2} \frac{\left[ \|a_n - x\|_2^2 - |u(x) - b_n|^2 \right]}{\left[ \left( u(x) - b_n \right) - \langle v, x - a_n \rangle \right]}$ *for $n \neq m$ and $r_m \stackrel{def}{=} +\infty$. Here in the definition of $r$ the $\min$ is taken only over $n$ for which $r_n > 0$ and $r_n \geq b_n - u_m(x)$. Also, for $c \in \mathbb{R}$ we define $c/0 = +\infty$.*

The condition on $a_n, b_n$ is imposed to avoid inconvenient cases when the center of a funnel lies on some another funnel. In practice, we compute the transport rays for a batch of points $x$ with tensor operations. We provide an example of transport rays of a random MinFunnel $u$ in Figure 9.

Part 3. **Defining the ray monotone map.** Let $\mathbb{P}$ be a distribution on $\mathbb{R}^D$ and $u$ be a MinFunnel. We aim to construct a $u$-ray monotone map $T : \mathbb{R}^D \to \mathbb{R}^D$ and define a distribution $\mathbb{Q} = T \sharp \mathbb{P}$.

If $\nabla u(x)$ does not exist, we define $T(x) = x$ (the set of such points is $\mu_L$-negligible). Otherwise, since $u$ is a MinFunnel, we have $\|\nabla u(x)\|_2 = 1$. Thus, the $u$-ray forward map $T$ may take any value $T(x) \in [x_0, x]$, where $x_0$ is the left (lower) endpoint of $ray(x) = [x_0, x_1]$. According to the analysis in §3.1, any (measurable) map $T$ defined by this principle is $u$-ray monotone. For such a map $T$ one may put $\mathbb{Q} = T \sharp \mathbb{P}$. As a result, for the pair $(\mathbb{P}, \mathbb{Q})$, function $T$ is an OT map (Corollary 1), possibly non-unique. Thus, the ground truth $\mathbb{W}_1(\mathbb{P}, \mathbb{Q})$ can be estimated from samples $x \sim \mathbb{P}$.

As we are also interested in testing how well OT dual solvers recover the OT **gradient** $\nabla f^*$, we need this gradient to exist and be $\mathbb{P}$-*unique*. From Proposition 1 we know that $f^* = u$ is an optimal potential. However, it is not necessarily unique (up to a constant). For example, in the trivial case $T(x) \equiv x$ and $\mathbb{P} = \mathbb{Q}$, *any* 1-Lipschitz $f^*$ is optimal and $\nabla f^*$ is not unique. We show that with mild assumptions on $\mathbb{P}, T, u$, the gradient can be designed to be unique and match $\nabla u(x)$.

**Proposition 4** (Uniqueness of the OT gradient). *Let $\mathbb{P}$ be absolutely continuous and let $u : \mathbb{R}^D \to \mathbb{R}$ be 1-Lipschitz. Let $T$ be a $u$-ray-monotone map for which $T(x) \neq x$ holds $\mathbb{P}$-almost surely, i.e., it moves every piece of mass of $\mathbb{P}$. Define $\mathbb{Q} = T \sharp \mathbb{P}$. Then for every optimal $f^*$ which maximizes* (5) *the equality $\nabla f^*(x) = \nabla u(x)$ holds $\mathbb{P}$-almost surely, i.e., the OT gradient $\nabla u$ is $\mathbb{P}$-unique.*

### 3.3 Benchmark Pairs

We use our methodology (§3.2) to construct **high-dimensional** ($D = 2, 2^2, \ldots, 2^7$) benchmark pairs and benchmark pairs on the space of $64 \times 64$ RGB **images** of celebrity faces ($D = 12288$).

For convenience, we use absolutely continuous $\mathbb{P}$ supported on a hypercube $\mathcal{S} = [-B, B]^D \subset \mathbb{R}^D$. In particular, for points $x \in \mathcal{S}$, we consider transport rays *truncated* to $\mathcal{S}$, i.e., $ray(x) \cap \mathcal{S}$. We

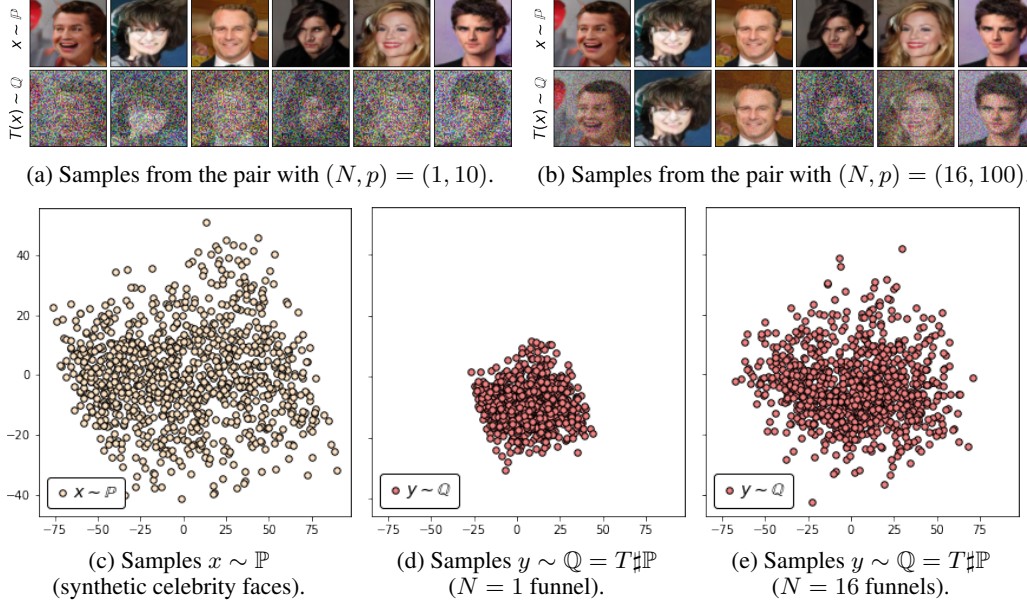

(a) Samples from the pair with $(N, p) = (1, 10)$.     (b) Samples from the pair with $(N, p) = (16, 100)$.

(c) Samples $x \sim \mathbb{P}$
(synthetic celebrity faces).

(d) Samples $y \sim \mathbb{Q} = T\sharp\mathbb{P}$
($N = 1$ funnel).

(e) Samples $y \sim \mathbb{Q} = T\sharp\mathbb{P}$
($N = 16$ funnels).

Figure 5: Visualization of random samples from our constructed Celeba images benchmark pairs.
In the last three plots, we show distributions projected to 2 principal components of $\mathbb{P}$.

construct the $u$-ray monotone map $T : \mathbb{R}^D \to \mathbb{R}^D$ by the following principle. Let $[x_0, x_1] \subset \mathcal{S}$ be the truncated transport ray of $u$ for $x \in \mathcal{S}$. We pick a parameter $p > 1$ and define the map $T$ as follows:

$$T(x) \stackrel{def}{=} \left[\frac{\|x - x_0\|_2}{\|x_0 - x_1\|_2}\right]^p x_0 + \left(1 - \left[\frac{\|x - x_0\|_2}{\|x_0 - x_1\|_2}\right]^p\right) x_1. \tag{12}$$

This is a power function, i.e., if a ray is parametrized as $[0, 1]$, it moves the mass along the ray by $t \mapsto t^p$. Since $\mathbb{P}$-almost all the points in $\mathcal{S}$ belong to (truncated) rays, we have $T(x) \neq x$ on $\mathcal{S}$.

**High-dimensional benchmark pairs.** In dimensions $D = 2, 2^2, \ldots, 2^7$, we put $\mathbb{P}$ to be the standard uniform distribution on $\mathcal{S} = [-2.5, 2.5]^D$. In each dimension $D$, we consider $N = 4, 16, 64, 256$ funnels and pick random parameters $a_n \sim \text{Uniform}([-2.5, 2.5]^D)$ and $b_n \sim \mathcal{N}(0, 0.1)$. For this initialization of $a_n, b_n$, the assumptions of Proposition 3 hold with probability 1. The random seed is hardcoded. We use $p = 8$. For the case $D = 2$, $N = 4$, we visualize the input distribution $\mathbb{P}$, the function $u$, the constructed map $T$ and the output distribution $\mathbb{Q} = T\sharp\mathbb{P}$ in Figure 3. We show examples of constructed $(\mathbb{P}, \mathbb{Q})$ for higher dimensions in Figure 4.

**Images benchmark pairs** for CIFAR-10 and Celeba. As $\mathbb{P}$ we consider the synthetic distributions of generated $32 \times 32$ RGB images ($D = 3072$) and $64 \times 64$ RGB images ($D = 12288$). To generate these images, we use the WGAN-QC [28] generator model trained on CIFAR-10 [26] and Celeba [30] datasets, respectively. For CIFAR-10, we train WGAN-QC by using its publicly available code.[2] For Celeba, we pick a readily available pre-trained generator from the related **Wasserstein-2 benchmark**[3] [23, §4.1]. To make $\mathbb{P}$ absolutely continuous, we add the Gaussian noise with axis-wise $\sigma = 0.01$. Then we truncate the distribution to $\mathcal{S} = [-1.1, 1.1]^D$, i.e., we reject samples which are out of $\mathcal{S}$. We construct two benchmark pairs per each dataset (Figures 5, 6) with $(N, p) = (1, 10), (16, 100)$ and $a_n \sim \text{Uniform}([-1., 1.]^D)$, $b_n \sim \mathcal{N}(0, 0.1)$.

In both high-dimensional and images pairs $(\mathbb{P}, \mathbb{Q})$, the input $\mathbb{P}$ is an absolutely continuous distribution supported on a hypercube $\mathcal{S}$. By the design, the distribution $\mathbb{P}$ is accessible by random samples. To sample from $\mathbb{Q}$, we first sample $x \sim \mathbb{P}$, then compute its transport ray (Proposition 3), truncate it to $\mathcal{S}$, and produce $y = T(x) \sim \mathbb{Q}$ by (12). The sampler for $\mathbb{Q}$ outputs only $y$, i.e., information about $u$, $T$, $x$ is hidden from the user and employed only when estimating the ground truth $\mathbb{W}_1(\mathbb{P}, \mathbb{Q})$ or $\nabla u$.

---

[2]`https://github.com/harryliew/WGAN-QC`
[3]`https://github.com/iamalexkorotin/Wasserstein2Benchmark`

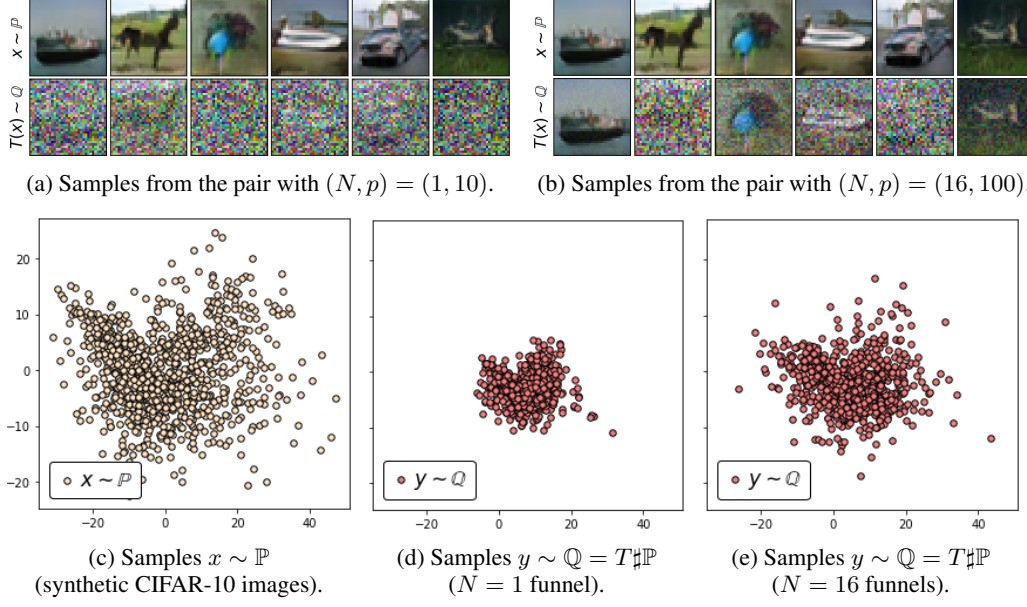

(a) Samples from the pair with $(N, p) = (1, 10)$.  (b) Samples from the pair with $(N, p) = (16, 100)$.

(c) Samples $x \sim \mathbb{P}$
(synthetic CIFAR-10 images).

(d) Samples $y \sim \mathbb{Q} = T\sharp\mathbb{P}$
($N = 1$ funnel).

(e) Samples $y \sim \mathbb{Q} = T\sharp\mathbb{P}$
($N = 16$ funnels).

Figure 6: Visualization of random samples from our constructed CIFAR-10 images benchmark pairs. In the last three plots, we show distributions projected to 2 principal components of $\mathbb{P}$.

**Reversed pairs.** Doing preliminary tests, we found that for solvers it is more challenging to compute the OT gradient for pair $(\mathbb{Q}, \mathbb{P})$ rather than $(\mathbb{P}, \mathbb{Q})$. In particular, in our images benchmark, the pair $(\mathbb{Q}, \mathbb{P})$ reflects the practical WGAN scenario better. Indeed, in WGANs, the solvers move the generated distribution (bad images, $\mathbb{Q}$ in our construction) to the real distribution (good images, $\mathbb{P}$). Recall that $\mathbb{Q} = T\sharp\mathbb{P}$, where $T$ is a differentiable bijection (12) along the (truncated) transport rays of a MinFunnel $u$. As a result, $\mathbb{Q}$ is absolutely continuous and the OT gradient for the reverse $(\mathbb{Q}, \mathbb{P})$ is $\mathbb{Q}$-unique (Proposition 4). For this pair, the optimal potential is $-u$ and its gradient is $-\nabla u$. To conclude, in all the experiments, we feed the **reverse pair** $(\mathbb{P}, \mathbb{Q}) := (\mathbb{Q}, \mathbb{P})$ to OT solvers in view.

## 4 Evaluation of Dual Solvers and Discussion

In this section, we evaluate WGAN dual OT solvers (§2) on our constructed benchmark (§3). Technical details of the implemetation are given in Appendix B. The code is written in PyTorch framework and is publicly available together with all the constructed benchmark distributions at



`https://github.com/justkolesov/Wasserstein1Benchmark`



**Metrics.** For $\mathbb{W}_1$, we do not use any specific metric but simply report obtained $\widehat{\mathbb{W}}_1$ and the ground truth $\mathbb{W}_1$. To quantify the recovered gradient $\nabla \hat{f}$, we use $\mathcal{L}^2$ and cosine similarity metrics [23, §4.2]:

$$
\mathcal{L}^2(\nabla \hat{f}, \nabla f^*) \overset{def}{=} \|\nabla \hat{f} - \nabla f^*\|_{\mathcal{L}^2}^2; \qquad \cos(\nabla \hat{f}, \nabla f^*) \overset{def}{=} \frac{\langle \nabla \hat{f}, \nabla f^* \rangle_{\mathcal{L}^2}}{\|\nabla \hat{f}\|_{\mathcal{L}^2} \cdot \|\nabla f^*\|_{\mathcal{L}^2}}, \qquad (13)
$$

where $\langle \nabla f_1, \nabla f_2 \rangle_{\mathcal{L}^2} \overset{def}{=} \int \langle \nabla f_1(x), \nabla f_2(x) \rangle d\mathbb{P}(x)$, $\|\nabla f\|_{\mathcal{L}^2}^2 \overset{def}{=} \langle \nabla f_1, \nabla f_2 \rangle_{\mathcal{L}^2}$ and $\nabla f^*$ is the ground truth OT gradient. The $\mathcal{L}^2$ metric compares the gradients $\nabla \hat{f}, \nabla f^*$ as elements of the space $\mathcal{L}^2(\mathbb{P})$ of quadratically integrable w.r.t. $\mathbb{P}$ functions. The cosine compares their directions *regardless* of the magnitude (Figure 13). To estimate the metrics, we use $2^{13}$ samples $x \sim \mathbb{P}$.

$\lfloor$**DOT**$\rceil$ For completeness, we add the *empirical* (batched) OT [9, 10, 36, 37] to evaluation. For batches $X \sim \mathbb{P}, Y \sim \mathbb{Q}$, we use $\widehat{\mathbb{W}}_1(X, Y)$ computed by a **discrete** solver as an estimate of $\mathbb{W}_1(\mathbb{P}, \mathbb{Q})$. The solver can be combined with the automatic differentiation to approximate $\nabla f^*$ on the batch $X$.

**High-dimensional pairs**. We test the solvers and report the estimated $\mathbb{W}_1$ value and metrics (13) in Tables 9, 10, 11 (Appendix C). We use fully-connected nets as potentials $f_\theta, g_\omega$ and movers $T_\theta, H_\omega$ (in the maximin solvers). In Figure 7, we show the potentials learned by solvers for $D = 2, N = 4$.

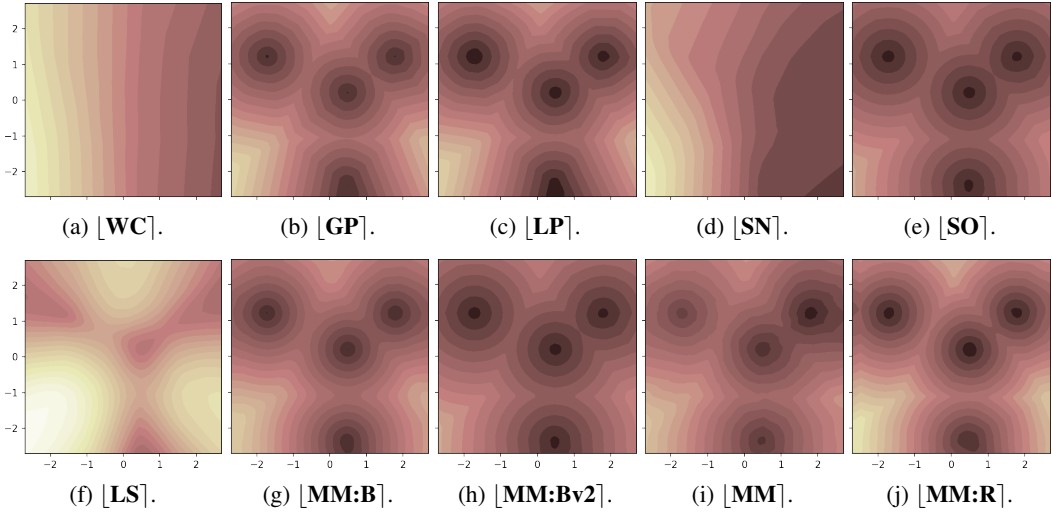

Figure 7: Surfaces of potentials $f_\theta$ learned by OT solvers on the pair with $D = 2, N = N$. For $\lceil \mathbf{MM:R} \rceil$ we plot (minus) second potential $-g_\omega$ as the solver does not compute the first one. The ground truth optimal potential is in Figure 3b.

As Table 11 shows, all the solvers randomly over/underestimate $\mathbb{W}_1$, typically with notable error. Consequently, *none of the solvers can be viewed as precise estimators of the OT cost*. Thus, our discussion below primarily focuses on the estimation of the OT gradient needed in WGANs.

Original $\lfloor \mathbf{WC} \rfloor$ leads to a pathological value surface of the learned potential (Figure 7a), which is far from the ground truth. This agrees with the claims of [17, §3]. Despite this, even in high dimensions the cos metric is positive (Table 9a), i.e., the recovered gradient correlates with the OT gradient. Popular $\lfloor \mathbf{GP} \rfloor$ notably improves upon $\lfloor \mathbf{WC} \rfloor$ and provides higher cos values (Table 9b). According to the results, its modification $\lfloor \mathbf{LP} \rfloor$ does not provide any major improvement (Table 9c).

Surprisingly, $\lfloor \mathbf{SN} \rfloor$ performs worse than $\lfloor \mathbf{GP} \rfloor$ and is comparable to $\lfloor \mathbf{WC} \rfloor$ in cos metric (Table 9d). We presume that this happens since the spectral norm negatively affects the expressive power of the neural net [1]. Even when $D = 2$ (Figure 7d), $\lfloor \mathbf{SN} \rfloor$ fails to recover the optimal potential. $\lfloor \mathbf{SO} \rfloor$ replaces spectral norm with orthogonalization and uses GroupSort activations which improve performance. The solver scores higher cos values (Table 9e) which are comparable to $\lfloor \mathbf{GP} \rfloor$.

Solver $\lfloor \mathbf{LS} \rfloor$ recovers biased potential since (7) is the duality formula for regularized OT which yields biased optimal potentials (Table 9f). The bias is huge for large $D, N$ when the benchmark pairs are complex. This is analogous to results for the same solver in the Wasserstein-2 ($\mathbb{W}_2$) benchmark [23].

Batch-based $\lfloor \mathbf{MM:B} \rfloor$ suffers from the bias in high dimensions (Table 9g) due to the overestimation of the value of the inner problem in (8). Note that cos metric values are even negative in high dimensions. The same is reported in the $\mathbb{W}_2$ benchmark [23]. $\lfloor \mathbf{MM\text{-}Bv2} \rfloor$ uses a more tricky optimization scheme and yields high cos values (Table 9h). It captures the direction of the OT gradient but *extremely* overestimates its magnitude, see high values of $\mathcal{L}^2$ in Table 10h.

Maximin solvers $\lfloor \mathbf{MM} \rfloor$, $\lfloor \mathbf{MM:R} \rfloor$ perform comparably to $\lfloor \mathbf{GP} \rfloor$, $\lfloor \mathbf{SO} \rfloor$, $\lfloor \mathbf{LP} \rfloor$, see Tables 9i, 9j. However, their training takes longer and sometimes diverges as it solves a saddle point problem. This agrees with the $\mathbb{W}_2$ benchmark [23, §4.3]. Using these solvers in GANs is not easy as it yields a challenging *min-max-min* optimization problem. Importantly, $\lfloor \mathbf{MM:R} \rfloor$ recovers the OT map which can itself be used as a generative model (*outside* the context of GANs) in computer vision tasks. Here we refer the reader to the recent *neural optimal transport* (NOT) methods [25, 24, 4, 22, 11, 44, 8].

Empirical $\lfloor \mathbf{DOT} \rfloor$ provides precise estimates of $\mathbb{W}_1$ (Table 11k) and the OT gradient (Table 9k) only in small dimensions. In high dimensions, it intolerably overestimates $\mathbb{W}_1$; its gradient is almost orthogonal to the ground truth ($\cos \lesssim 0$). This is due to the exponential (in $D$) sample complexity of DOT [53]. Thus, (unregularized) DOT is an imprecise estimator of $\mathbb{W}_1$ or the OT gradient in high $D$.

**Images pairs.** Here we do not consider $\lfloor \mathbf{SO} \rfloor$ as its authors do not provide convolutional architectures with GroupSort and orthonormalization. We use DCGAN [42] as potentials $f_\theta, g_\omega$. In $\lfloor \mathbf{MM} \rfloor$ and $\lfloor \mathbf{MM:R} \rfloor$, the movers $T_\theta, H_\omega$ are UNets [43]. The evaluation results on Celeba benchmark pairs are given in Tables 6, 7, 8 and on CIFAR-10 benchmark pairs – in Tables 3, 4, 5 (Appendix C).

| | | $\lfloor$**WC**$\rceil$ | $\lfloor$**GP**$\rceil$ | $\lfloor$**LP**$\rceil$ | $\lfloor$**SN**$\rceil$ | $\lfloor$**SO**$\rceil$ | $\lfloor$**LS**$\rceil$ | $\lfloor$**MM:B**$\rceil$ | $\lfloor$**MM:Bv2**$\rceil$ | $\lfloor$**MM**$\rceil$ | $\lfloor$**MM:R**$\rceil$ | $\lfloor$**DOT**$\rceil$ |
|---|---|---|---|---|---|---|---|---|---|---|---|---|
| $cos$ | **HD** | 🙁 | 🙂 | 🙂 | 🙁 | 🙂 | 🙁 | 😐 | 😐 | 😐 | 🙂 | 🙁 |
| | **IMG** | 😐 | 🙂 | 🙂 | 😐 | - | 😐 | 😐 | 😐 | 🙁 | 🙂 | 🙂 |
| $\mathcal{L}^2$ | **HD** | 🙁 | 😐 | 🙂 | 🙁 | 😐 | 🙁 | 😐 | 😐 | 🙁 | 😐 | 🙁 |
| | **IMG** | 🙁 | 🙁 | 🙁 | 🙁 | - | 🙁 | 🙁 | 🙁 | 🙁 | 🙂 | 🙁 |
| $\mathbb{W}_1$ | **HD** | 🙁 | 😐 | 🙂 | 🙁 | 😐 | 🙁 | 🙁 | 😐 | 😐 | 😐 | 🙁 |
| | **IMG** | 🙁 | 🙁 | 🙁 | 🙁 | - | 😐 | 🙁 | 😐 | 😐 | 😐 | 🙂 |

Table 1: The summary of WGAN dual OT solvers' performance in $cos$, $\mathcal{L}^2$ and $\mathbb{W}_1$ metrics on our high-dimensional (HD) and images (IMG) benchmark pairs. For details, see Appendix C.

Surprisingly, our images benchmark pairs turned to be simpler than some high-dimensional benchmark pairs. Although $(\mathbb{P}, \mathbb{Q})$ are absolutely continuous and supported on $[-1.1, 1.1]^D$, their actual probability mass is still concentrated around small low-dimensional sub-manifold of data. We suppose that this is one of the causes for the reasonable performance of most solvers. In particular, we see that even $\lfloor$**DOT**$\rceil$, $\lfloor$**LS**$\rceil$, $\lfloor$**MM:B**$\rceil$ score $cos > 0$ in images benchmark pairs, although they struggled to produce good results on high-dimensional pairs.

Solvers $\lfloor$**GP**$\rceil$, $\lfloor$**LP**$\rceil$, $\lfloor$**MM:R**$\rceil$ provide very high $cos > 0.9$ (Table 6). However, only $\lfloor$**MM:R**$\rceil$ provides precise approximation of $\nabla f^*$ in $\mathcal{L}^2$ norm (Table 7). Interestingly, maximin $\lfloor$**MM**$\rceil$ diverges on our images benchmark pairs (tuning the hyper-parameters did not help). Similar to the evaluation in high-dimensional pairs, $\lfloor$**WC**$\rceil$ and $\lfloor$**SN**$\rceil$ show moderate $cos > 0$, but its value is notably smaller than that of the top-performing methods. Solver $\lfloor$**MM:Bv2**$\rceil$ demonstrates meaningful estimate of $\mathbb{W}_1$ (Table 8). Nevertheless, its recovered gradient is almost orthogonal to the ground truth ($cos \approx 0$, see Table 6), and the values of $\mathcal{L}^2$ metric are **extremely** high (Table 7).

## 5 Discussion

Our methodology creates pairs of continuous distributions with known ground truth OT cost and gradient, filling the missing gap of benchmarking $\mathbb{W}_1$ dual solvers. This development allows us to evaluate the OT performance of WGAN dual methods. The experimental results are summarized in Table 1. Our evaluation shows that these solvers should **not** be considered as **meaningful estimators** of $\mathbb{W}_1$ as they exhibit large error. However, the OT gradient recovered by these solvers shows positive $cos$ with ground truth. This suggests that most methods could still be used to **minimize** $\mathbb{W}_1$ in variational problems, e.g., Wasserstein GANs.

**Computational complexity.** Evaluation of all the OT solvers on our high-dimensional and images benchmark pairs takes less than 50 hours on a single GPU GTX 1080ti (11 GB VRAM).

**Potential Impact.** Our benchmark distributions can be used to evaluate future dual OT solvers in high-dimensional spaces, a crucial step to improve the transparency and replicability of OT and WGAN-related research. We expect our benchmark to become a standard benchmark for $\mathbb{W}_1$ optimal transport as part of the ongoing effort of advancing computational OT.

**Limitations (benchmark).** We rely on MinFunnels as optimal Kantorovich potentials to generate benchmark pairs. Also, we limit our pairs to be absolutely continuous distributions. It is unclear whether our benchmark sufficiently reflects the real-world scenarios in which the WGAN solvers are used. Nevertheless, our methodology is generic and can be used to construct new benchmark pairs.

**Limitations (evaluation).** We evaluate how well the OT solvers compute OT cost and gradient but do not assess their performance in GAN settings. Studying this question is a promising future research avenue which could help to develop new OT-based methods for generative modeling.

ACKNOWLEDGEMENTS. The work was supported by the Analytical center under the RF Government (subsidy agreement 000000D730321P5Q0002, Grant No. 70-2021-00145 02.11.2021).

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
