## A  Proofs

*Proof of Proposition 1.* For all $x, y \in \mathbb{R}^D$, it holds $u(x) - u(y) \leq \|u(x) - u(y)\|_2 \leq \|x - y\|_2$ since $u$ is 1-Lipschitz. Let $\pi^*$ be an OT plan. We compute

$$\mathbb{W}_1(\mathbb{P}, \mathbb{Q}) = \int_{\mathbb{R}^D \times \mathbb{R}^D} \|x - y\|_2 d\pi^*(x, y) \geq \int_{\mathbb{R}^D \times \mathbb{R}^D} \big(u(x) - u(y)\big) d\pi^*(x, y) = \quad (14)$$

$$\int u(x) d\mathbb{P}(x) - \int u(x) d\mathbb{Q}(y) = \int_{\mathbb{R}^D \times \mathbb{R}^D} \big(u(x) - u(y)\big) d\pi(x, y) = \int_{\mathbb{R}^D \times \mathbb{R}^D} \|x - y\|_2 d\pi(x, y), \quad (15)$$

where in line (15), we use the fact that $u(x) - u(y) = \|x - y\|_2$ holds $\pi$-almost surely for all $x, y \in \mathbb{R}^D$. As a result, the transport cost of $\pi$ is not greater than the OT cost and $\pi$ is an OT plan. Thus, (14) is an equality, and $u$ attains the maximum in (5), i.e., it is an optimal dual potential. $\square$

*Proof of Proposition 2.* Since $\mathcal{S} \subset \mathbb{R}^D$ is a compact set, there exists a finite $\frac{\epsilon}{2}$-coverage, i.e., a set of points $a_n \in \mathbb{R}^D$ ($n = 1, 2, \ldots, N$) such that for all $x$ there exists $n(x)$ with $\|x - a_{n(x)}\|_2 \leq \frac{1}{2}\epsilon$. We put $b_n = f^*(a_n)$ and consider the MinFunnel $u$ (11) with parameters $\{a_n, b_n\}_{n=1}^N$.

Now we pick any $x \in \mathcal{S}$ and show that $|u(x) - f^*(x)| \leq \epsilon$. First, we note that

$$u(x) = \min_n \{\|x - a_n\|_2 + b_n\} \leq \|x - a_{n(x)}\|_2 + b_n \leq \frac{1}{2}\epsilon + f^*(a_{n(x)}) \leq$$

$$\frac{1}{2}\epsilon + f^*(x) + \|f^*(a_{n(x)}) - f^*(x)\|_2 \leq \frac{1}{2}\epsilon + f^*(x) + \|a_{n(x)} - x\|_2 = f^*(x) + \epsilon, \quad (16)$$

where we use the fact that $f^*$ is 1-Lipschitz continuous. Now we note that for every $n$ it holds:

$$f^*(x) \leq f^*(a_n) + \|x - a_n\|_2 = b_n + \|x - a_n\|_2. \quad (17)$$

By taking $\min$ over all $n$ in (17), we get that $f^*(x) \leq u(x)$ for all $x$. By combining this fact with (16), we see that $u(x) \in [f^*(x), f^*(x) + \epsilon]$ for all $x \in \mathcal{S}$, i.e., $\sup_{x \in \mathcal{S}} |f^*(x) - u(x)| \leq \epsilon$. $\square$

Proposition 2 yields that MinFunnels (11) can approximate any 1-Lipschitz optimal potential $f^*$ in (5) in supremum norm. Be careful as this does necessarily guarantee that for $f^*$-ray monotone map $T^* : \mathbb{R}^D \to \mathbb{R}^D$, there exists a MinFunnel $u$ (approximating $f^*$) for which one may construct $u$-ray monotone map $T$ approximating $T^*$. We do not know if this holds for MinFunnel functions. However, this is indiffirent for us as we do not aim to approximate a specific OT map $T^*$ but construct a random one (§3.3).

*Proof of Proposition 3.* We split the proof into 4 parts.

**Part 1** [Uniqueness of $m$.] We show that $\arg\min_n \{u_n(x)\}$ contains only one index, i.e., $m$ is uniquely defined. Assume the contrary, i.e., that there exist $m \neq m'$ such that at $x$ we have

$$u(x) = u_m(x) = \|x - a_m\|_2 + b_m = u_{m'}(x) = \|x - a_{m'}\|_2 + b_{m'} = \min_n \{u_n(x)\}.$$

Since $u$ is 1-Lipschitz, we have $u(a_m) \geq u(x) - \|x - a_m\|_2 = u_m(x) - \|x - a_m\|_2 = b_m$. On the other hand, $u(a_m) = \min_n\{u_n(a_m)\} \leq u_m(a_m) = \|a_m - a_m\| + b_m = b_m$. We combine these two inequalities and obtain $u(a_m) = u_m(a_m) = b_m$. This means that

$$u(x) - u(a_m) = u_m(x) - u_m(a_m) = \|x - a_m\|_2. \quad (18)$$

Consequently, $u$ is affine on the segment $[a_m, x]$, see [47, Lemma 3.5]. Analogously, we get that $u(a_{m'}) = u_{m'}(a_{m'}) = b_{m'}$ and $u$ is affine on the segment $[a_{m'}, x]$. By the assumption of the proposition, $u$ is differentiable at $x$. As a result, we obtain $\nabla u(x) = \frac{x - a_m}{\|x - a_m\|_2} = \frac{x - a_{m'}}{\|x - a_{m'}\|_2}$. This yields that vectors $x - a_{m'}$ and $x - a_m$ are collinear, i.e., $a_{m'} \in [x, a_m]$ or $a_m \in [x, a_{m'}]$. Without loss of generality, we consider the first case. We have $u(a_{m'}) = b_{m'} = \|a_{m'} - a_m\|_2 + b_m$. This is a contradiction since $\|a_{m'} - a_m\|_2 \neq |b_m - b_{m'}|$ by the assumption of the proposition. We conclude that $\arg\min_n \{u_n(x)\}$ contains only one index $m$. In particular, we see that $x$ is not an intersection point of funnels $u_n$. Also, $x$ is not a center $a_m$ as a funnel is not differentiable at its center.

**Part 2** [Direction of ray$(x)$.] Since $u$ is 1-Lipschitz, we have $u(a_m) \geq u(x) - \|a_m - x\|_2 = u_m(x) - \|a_m - x\|_2 = b_m$. On the other hand, $u(a_m) = \min_n\{u_n(a_m)\} \leq u_m(a_m) = \|a_m - a_m\|_2 + b_m$. Thus, $u(a_m) = u_m(a_m) = b_m$, equation (18) holds, $u$ is affine on $[x, a_m]$ and $\nabla u(x) = \nabla u_m(x) = \frac{x - a_m}{\|x - a_m\|_2}$. Thus, the direction of a transport ray of $x$ is given by $v \overset{def}{=} \nabla u(x) = \frac{x - a_m}{\|x - a_m\|_2}$. By the definition of a transport ray (§3.1), we conclude that $[a_m, x] \subset \mathrm{ray}(x)$.

**Part 3** [Lower endpoint of ray$(x)$.] We prove that $a_m$ is the lower endpoint. Assume that there exists $x_0 \neq a_m$ such that $a_m \in [x_0, x]$ and $u(x_0) + \|a_m - x_0\|_2 = u(a_m) = u_m(a_m) = b_m$. Consider any $m' \in \arg\min_n\{u_n(x_0)\}$, i.e., $u(x_0) = u_{m'}(x_0) = \|x_0 - a_{m'}\|_2 + b_{m'}$. Note that $m \neq m'$ since

$$u(x_0) = u(a_m) - \underbrace{\|x_0 - a_m\|_2}_{>0} < u(a_m) = u_m(a_m) = \min_{x' \in \mathbb{R}^D} u_m(x').$$

We are going to prove that $a_m \in [a_{m'}, x]$. Again, we note that due to 1-Lipschitz continuity of $u$, we have $u(a_{m'}) \geq u(x_0) - \|x_0 - a_{m'}\|^2 = u_{m'}(x_0) - \|x_0 - a_{m'}\|^2 = b_{m'}$. On the other hand, $u(a_{m'}) = \min_n\{u_n(a_n)\} \leq u_{m'}(a_{m'}) = b_{m'}$. Thus, $u(a_{m'}) = u_{m'}(a_{m'}) = b_{m'}$. We have

$$u(x_0) - u(a_{m'}) = u_{m'}(x_0) - u_{m'}(a_{m'}) = \|x_0 - a_{m'}\|_2.$$

We also know that $u(a_m) - u(x_0) = u_m(a_m) - u_m(x_0) = \|a_m - x_0\|_2$. By summing these equalities, we get $u(a_m) - u(a_{m'}) = \|a_m - a_{m'}\|_2$. On the other hand, the same quantity equals $b_m - b_{m'}$. Thus, $\|a_m - a_{m'}\|_2 = |b_m - b_{m'}|$, which is a contradiction to the assumption of the proposition.

**Part 4** [Upper endpoint of ray$(x)$.] The upper endpoint is a point $x + rv$ such that $r$ is the maximal non-negative value for which $u(x + rv) = u(x) + r$. Note that $x + rv$ is the point where $u_m = u_n$ for some $n \neq m$. If there is no intersection, the ray is infinite and $r = +\infty$. Let us find where $u_m$ equals $u_n$ ($m \neq n$) on the ray $x + r_n v$ ($r_n > 0$). We need to find $r_n > 0$ by solving

$$u_m(x + r_n v) = u_m(x) + r_n = u_n(x + r_n v) = \|r_n v + x - a_n\|_2 + b_n,$$

or, equivalently,

$$u_m(x) + (r_n - b_n) = \|r_n v + x - a_n\|_2. \tag{19}$$

The left side must be non-negative, i.e., $r_n \geq b_n - u_m(x)$. We take the square of both sides:

$$u_m^2(x) + (r_n - b_n)^2 + 2(r_n - b_n)u_m(x) = \|x - a_n\|_2^2 + r_n^2 + 2r\langle x - a_n, v\rangle.$$

This is a linear equation in $r_n$ as $r_n^2$ terms vanish. We derive

$$r_n \cdot \left[(u_m(x) - b_n) - \langle v, x - a_n\rangle\right] = \frac{1}{2}\left[\|a_n - x\|_2^2 - |u_m(x) - b_n|^2\right]. \tag{20}$$

Consider the case when the right side is **zero**, i.e., $\|a_n - x\|_2 = |u_m(x) - b_n|$. We know that $u_m(x) = u(x) < u_n(x) = \|x - a_n\|_2 + b_n$. Thus, $\|a_n - x\|_2 = b_n - u_m(x)$, and (20) equals

$$r_n \cdot \left[-\|x - a_n\|_2 - \langle v, x - a_n\rangle\right] = 0. \tag{21}$$

Recall that $\|v\|_2 = 1$. Thus, (21) may have a positive solution $r_n$ only when $x = a_n$ or $(x \neq a_n) \wedge \left(v = -\frac{x - a_n}{\|x - a_n\|_2}\right)$. In the **first** case, $u_n(x) = u_n(a_n) = b_n = u_m(x)$ which contradicts to $u_m(x) \neq u_n(x)$. Thus, this case is not possible. In the **second** case, (19) equals

$$r_n - \|x - a_n\|_2 = |1 - \frac{r_n}{\|x - a_n\|_2}| \cdot \|x - a_n\|_2.$$

Thus, all $r_n \geq \|a_n - x\|_2 = b_n - u_m(x)$ are the solutions. We pick $r_n = \|a_n - x\|_2$ and see that

$$u_m(x) + r_n = u_m(x + r_n v) = u_n(x + r_n v) = \|r_n v + x - a_n\|_2 + b_n = b_n,$$

i.e., the expression equals the lowest value $b_n$ of $u_n$. Thus, $x + r_n v = a_n$, i.e., it is the center of the funnel $u_n$. In particular, $x \in [a_m, a_n]$. We also derive that

$$u_m(x) + r_n = u_m(x) + \|x - a_n\|_2 = u_n(a_n) = b_n.$$

Recall that $u_m(x) = \|x - a_m\|_2 + b_m$. Thus, $\|x - a_m\|_2 + b_m + \|x - a_n\|_2 = b_n$. Since $x \in [a_m, a_n]$, we conclude that $\|a_n - a_m\|_2 + b_m = b_n$. This provides $\|a_n - a_m\|_2 = |b_n - b_m|$ which is a contradiction to the assumptions of the proposition. Thus, the second case is also not possible.

If the right side of (20) is **non-zero**, we derive

$$r_n = \frac{1}{2}\big[\|a_n - x\|_2^2 - |u(x) - b_n|^2\big] / \big[\big(u(x) - b_n\big) - \langle v, x - a_n\rangle\big]. \tag{22}$$

When the denominator is zero, we put $r_n = +\infty$, i.e., funnels $u_m, u_n$ do not intersect in any $x + r_n v$.

To conclude, the intersection with $u_n$ does not happen when $r_n < b_n - u_m(x)$. Otherwise, the intersection happens at a point $x + r_n v$, where $r_n$ is defined by (22) and equals $+\infty$ if denominator/numerator is zero (no intersection). The upper endpoint of ray$(x)$ is given by $x + rv$ with $r = \min r_n$ (the first intersection), where the $\min$ is taken over $r_n$ such that $r_n \geq b_n - u_m(x)$. $\square$

*Proof of Proposition 4.* We compute $\mathbb{W}_1$ by substituting $f^*$ to (5):

$$\mathbb{W}_1(\mathbb{P}, \mathbb{Q}) = \int f^*(x)d\mathbb{P}(x) - \int f^*(y)d\mathbb{Q}(y) = \int f^*(x)d\mathbb{P}(x) - \int f^*\big(T(x)\big)d\mathbb{P}(x) = \tag{23}$$

$$\int \big[f^*(x) - f^*\big(T(x)\big)\big]d\mathbb{P}(x) \leq \int \|x - T(x)\|_2 d\mathbb{P}(x) = \mathbb{W}_1(\mathbb{P}, \mathbb{Q}), \tag{24}$$

where in line (23), we use the change of variables for $y = T(x)$ and equality $T\sharp\mathbb{P} = \mathbb{Q}$. In line (24), we use the fact that $f^*$ is 1-Lipschitz. From lines (23) and (24), we conclude that $f^*(x) - f^*\big(T(x)\big) = \|x - T(x)\|_2$ holds $\mathbb{P}$-almost surely, $T$ is $f^*$-ray monotone and (24) is the equality. Recall that we have $T(x) \neq x$ by the assumption. Consequently (§3.1), $f^*$ is affine on a segment $[x, T(x)]$ which is contained in some transport ray of $f^*$. If $f^*$ is diffirentiable at $x$, then it necessarily holds

$$\nabla f^*(x) = \frac{x - T(x)}{\|x - T(x)\|_2}. \tag{25}$$

Since $\mathbb{P}$ is absolutely continuous, the set of points $x$ where $\nabla f^*(x)$ does not exist is $\mathbb{P}$-neblibigle. Therefore, (25) holds $\mathbb{P}$-almost surely. By conducting the same analysis for $u$, we derive that $\nabla u(x)$ also equals (25). Therefore, $\nabla u(x) = \nabla f^*(x)$ holds $\mathbb{P}$-almost surely. $\square$

# B  Implementation Details

In this section, we provide the details of the training of the OT solvers that we consider. The `PyTorch` source code to create the benchmark pairs and evaluate existing solvers is publicly available at

$$\texttt{https://github.com/justkolesov/Wasserstein1Benchmark}$$

## B.1  Neural Networks

**High-dimensional pairs.** We use multi-layer perceptrons (MLP) for potentials $f_\theta$ and $g_\omega$ (where applicable) with ReLU activations and the following hidden layer sizes:

$$[\max(2D, 128), \max(2D, 128), \max(D, 128)]. \tag{26}$$

Here $D$ is the dimension of the ambient space. In $\lfloor\textbf{SN}\rfloor$[4], we apply spectral normalization to weights of the linear layers by using the power iteration method. In $\lfloor\textbf{SO}\rfloor$[5], we use FullSort activations instead of ReLU. Besides, we enforce ortho-normality on weight matrices by using `geotorch.orthogonal`.[6] In maximin $\lfloor\textbf{MM}\rfloor$, $\lfloor\textbf{MM:R}\rfloor$, mover $T_\theta$ (or $H_\omega$) is a ReLU MLP with the same layer sizes as (26).

**Images pairs.** The potential $f_\theta$ (or $g_\omega$) has DCGAN[7] architecture without the batch normalization. In $\lfloor\textbf{SN}\rfloor$, we apply spectral normalization to weights of the linear layers by using the power iteration method. In maximin $\lfloor\textbf{MM}\rfloor$, $\lfloor\textbf{MM:R}\rfloor$, mover $T_\theta$ (or $H_\omega$) has UNet[8] architecture.

Note that we remove the last $\tanh$ layer which is used in DCGAN as it directly contradicts the Kantorovich's duality formula (5). The potential $f$ should not be bounded. For example, $\mathbb{P} = \text{Uniform}\big([-a, 0]\big)$ and $\mathbb{Q} = \text{Uniform}\big([0, a]\big)$, the optimal potential is $u(x) = -x$. It can not be learned by a net whose outputs are limited to $(-1, 1)$ when $|a| > 1$. In practice, we found that DCGAN **with** the output $\tanh$ really fails to learn anything meaningful on our benchmark ($\cos \approx 0$).

---

[4]`github.com/christiancosgrove/pytorch-spectral-normalization-gan`
[5]`github.com/cemanil/LNets`
[6]`github.com/Lezcano/geotorch`
[7]`pytorch.org/tutorials/beginner/dcgan_faces_tutorial.html`
[8]`github.com/milesial/Pytorch-UNet`

## B.2 Optimization

In all the cases, we use **Adam** optimizer [21] with $lr = 2 \cdot 10^{-4}$ and $\beta_1 = 0$, $\beta_2 = 0.9$. Working in high dimensions, we set the batch size to 1024. In the images case, the batch size is 32.

Doing preliminary experiments, we noted that even when the *train* loss (which the solver optimizes) converges, the *test* metrics, e.g., cosine similarity, still may continue improving. Thus, we optimize the methods until their *test* cos metric converges as well. The details are summarized in Table 2. Specifically for dimension $D = 2$, we increase the number of iterations for each solver 5 times.

| SOLVER | HIGH-DIMENSIONAL PAIRS | IMAGES PAIRS |
|---|---|---|
| ⌊**WC**⌉ | 5000 iters, c = 0.04 | 5000 iters, c = 0.04 |
| ⌊**GP**⌉ | 40000 iters, $\lambda_{GP} = 10$ | 20000 iters, $\lambda_{GP} = 10$ |
| ⌊**LP**⌉ | 40000 iters, $\lambda_{LP} = 10$ | 20000 iters, $\lambda_{LP} = 10$ |
| ⌊**SN**⌉ | 5000 iters, 5 power iterations [35] | 5000 iters, 5 power iterations |
| ⌊**SO**⌉ | 15000 iters, full sort | N / A |
| ⌊**LS**⌉ | 10000 iters, $\mathcal{L}^2$ reg., $\epsilon = 0.01$ [48, Eq. 7] | 10000 iters, $\mathcal{L}^2$ reg., $\epsilon = 0.01$ |
| ⌊**MM:B**⌉ | 15000 iters | 20000 iters |
| ⌊**MM:Bv2**⌉ | 15000 iters | 20000 iters |
| ⌊**MM**⌉ | 15000 iters, 12 steps of $H_\omega$ per 1 step of $f_\theta$. | 10000 iters, 12 steps of $H_\omega$ per 1 step of $f_\theta$. |
| ⌊**MM:R**⌉ | 15000 iters, 12 steps of $T_\theta$ per 1 step of $g_\omega$. | 10000 iters, 12 steps of $T_\theta$ per 1 step of $g_\omega$. |

Table 2: Hyperparameters of the OT solvers.

## C  Metrics

For all the experiments, we report the metric values obtained after a single random restart. We do not report the results of multiple random restarts. We found that they mostly provide the same observations which do not affect the general trends of performance reported in our paper.

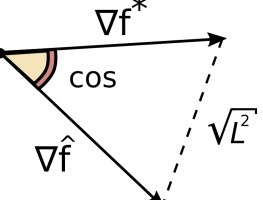

Figure 8: Cos and $\mathcal{L}^2$ show the angle and distance between $\nabla \hat{f}$ and $\nabla f^*$ in $\mathcal{L}^2(\mathbb{P})$.

In ⌈**DOT**⌋, we compute all the metrics by using $2^{12}$ random samples $X \sim \mathbb{P}$, $Y \sim \mathbb{Q}$. In most neural solvers, we estimate $\mathbb{W}_1(\mathbb{P}, \mathbb{Q})$ by using

$$\mathbb{W}_1(\mathbb{P}, \mathbb{Q}) \approx \frac{1}{|X|} \sum_{x \in X} f_\theta(x) - \frac{1}{|Y|} \sum_{y \in Y} f_\theta(y),$$

where $X \sim \mathbb{P}$, $Y \sim \mathbb{Q}$ are random batches of size $2^{13}$. For simplicity, we omit the regularization terms, e.g., the gradient penalty, lipschitz penalty, etc. In ⌈**LS**⌋ and ⌈**MM:R**⌋, we use

$$\mathbb{W}_1(\mathbb{P}, \mathbb{Q}) \approx \frac{1}{|X|} \sum_{x \in X} f_\theta(x) - \frac{1}{|Y|} \sum_{y \in Y} g_\omega(y), \qquad \mathbb{W}_1(\mathbb{P}, \mathbb{Q}) \approx -\frac{1}{|X|} \sum_{x \in X} g_\omega(x) + \frac{1}{|Y|} \sum_{y \in Y} g_\omega(y).$$

The results for high-dimensional and images pairs are given below.

**Technical remark**. Doing preliminary experiments in small dimension $D = 2$, we noted that most methods learn reasonable surfaces of the potentials (Figure 7) but struggle to provide high metrics, see, e.g., Table 9. The cause of this is a *numerical error*, see below.

By the construction of our benchmark, we get $y \sim \mathbb{Q}$ by moving $x \sim \mathbb{P}$ (the uniform distribution on hypercube $\mathcal{S}$) along its ray$(x) = [x_0, x_1]$ closer to the center $x_0 = a_n$ of the funnel $u_m$ (Proposition 3). We apply the function $t \mapsto t^p$ along the ray (12). In small dimensions, centers $a_n$ of funnels are sufficiently dense in the hypercube $\mathcal{S}$. As a result, points $x \sim \mathbb{P}$ with high probability are located next to centers of funnels, i.e., $\|x - x_0\|_2$ is small. In turn, $t = \|x - x_0\|_2 / \|x_0 - x_1\|_2$ is also small and $t^p \approx 0$ ($p > 1$), i.e., $y = T(x) \approx x_0$. Due to the numerical error of handling small numbers, $T(x)$ may output a point which is close to $x_0$ (the funnel center) but lies **not** on the ray $[x_0, x_1]$ or simply the point $x_0$ itself. This makes the evaluation of $\nabla u(y)$ problematic. In the first case, it may differ from $\nabla u(x)$ as $y$ fell out of the ray$(x)$. In the second case, $u$ is not even differentiable at $x_0 = a_n$.

The issue vanishes in higher dimensional pairs ($D \geq 4$) since the probability of a point $x \in \mathcal{S}$ to be close to the center of some funnel tends to zero. To fix the issue for $D = 2$, one may put smaller $p$ and generate new benchmark pairs. We use $p = 8$ in all high-dimensional pairs for consistency.

| N\Solver | $\lceil$WC$\rceil$ | $\lceil$GP$\rceil$ | $\lceil$LP$\rceil$ | $\lceil$SN$\rceil$ | $\lceil$LS$\rceil$ | $\lceil$MM:B$\rceil$ | $\lceil$MM:Bv2$\rceil$ | $\lceil$MM$\rceil$ | $\lceil$MM:R$\rceil$ | $\lceil$DOT$\rceil$ | True |
|---|---|---|---|---|---|---|---|---|---|---|---|
| 1 | 0.25 | 0.94 | 0.94 | 0.36 | 0.60 | 0.59 | 0.50 | 0.11 | 0.95 | 0.78 | 1.00 |
| 16 | 0.33 | 0.92 | 0.92 | 0.35 | 0.36 | 0.38 | 0.09 | 0. 46 | 0.91 | 0.55 | 1.00 |

Table 3: Cos↑ metric values of the OT gradient estimated by OT solvers on our **CIFAR-10 IMAGES** benchmark pairs $(\mathbb{P}, \mathbb{Q})$, dimension $D = 3072$. Colors indicate the metric value: $\cos > 0.85$, $\cos \in (0.5, 0.85]$, $\cos \in (0.15, 0.5]$, $\cos \leq 0.15$.

| N\Solver | $\lceil$WC$\rceil$ | $\lceil$GP$\rceil$ | $\lceil$LP$\rceil$ | $\lceil$SN$\rceil$ | $\lceil$LS$\rceil$ | $\lceil$MM:B$\rceil$ | $\lceil$MM:Bv2$\rceil$ | $\lceil$MM$\rceil$ | $\lceil$MM:R$\rceil$ | $\lceil$DOT$\rceil$ | True |
|---|---|---|---|---|---|---|---|---|---|---|---|
| 1 | ≫10 | 1.79 | 1.70 | 2.60 | 0.94 | 1.04 | 1.74 | ≫10 | 0.08 | 0.45 | 0.00 |
| 16 | ≫10 | 1.34 | 1.12 | 3.32 | 1.87 | 1.28 | ≫10 | ≫10 | 0.16 | 0.95 | 0.00 |

Table 4: $\mathcal{L}^2$↓ metric values of the OT gradient estimated by OT solvers on our **CIFAR-10 IMAGES** benchmark pairs $(\mathbb{P}, \mathbb{Q})$, dimension $D = 3072$. Colors indicate the metric value: $\mathcal{L}^2 < 0.25$, $\mathcal{L}^2 \in [0.25, 0.65)$, $\mathcal{L}^2 \in [0.65, 1.2)$, $\mathcal{L}^2 \geq 1.2$.

| N\Solver | WC | $\lceil$GP$\rceil$ | $\lceil$LP$\rceil$ | $\lceil$SN$\rceil$ | $\lceil$LS$\rceil$ | $\lceil$MM:B$\rceil$ | $\lceil$MM:Bv2$\rceil$ | $\lceil$MM$\rceil$ | $\lceil$MM:R$\rceil$ | $\lceil$DOT$\rceil$ | True |
|---|---|---|---|---|---|---|---|---|---|---|---|
| 1 | ≫100 | 66.25 | 66.79 | 20.44 | 17.00 | 15.90 | 35.26 | 35.10 | 24.29 | 31.16 | 29.66 |
| 16 | ≫100 | 36.16 | 36.75 | 13.26 | 5.54 | 4.21 | 24.36 | ≫100 | 23.46 | 25.34 | 18.82 |

Table 5: $\mathbb{W}_1$ metric values estimated by OT solvers on our **CIFAR-10 IMAGES** benchmark pairs $(\mathbb{P}, \mathbb{Q})$, dimension $D = 3072$. Colors indicate the value of the relative deviation of $\widehat{\mathbb{W}}_1$ from $\mathbb{W}_1$, i.e., dev $\stackrel{def}{=} 100\% \cdot \frac{|\mathbb{W}_1 - \widehat{\mathbb{W}}_1|}{\mathbb{W}_1}$: dev $< 15\%$, dev $\in [15, 30\%)$, dev $\in [30, 50)\%$, dev $\geq 50\%$

| N\Solver | $\lceil$WC$\rceil$ | $\lceil$GP$\rceil$ | $\lceil$LP$\rceil$ | $\lceil$SN$\rceil$ | $\lceil$LS$\rceil$ | $\lceil$MM:B$\rceil$ | $\lceil$MM:Bv2$\rceil$ | $\lceil$MM$\rceil$ | $\lceil$MM:R$\rceil$ | $\lceil$DOT$\rceil$ | True |
|---|---|---|---|---|---|---|---|---|---|---|---|
| 1 | 0.35 | 0.96 | 0.95 | 0.36 | 0.43 | 0.43 | 0.32 | 0.01 | 0.97 | 0.64 | 1.00 |
| 16 | 0.48 | 0.92 | 0.92 | 0.42 | 0.11 | 0.14 | 0.01 | 0.20 | 0.92 | 0.25 | 1.00 |

Table 6: Cos↑ metric values of the OT gradient estimated by OT solvers on our **CELEBA IMAGES** benchmark pairs $(\mathbb{P}, \mathbb{Q})$, dimension $D = 12288$. Colors indicate the metric value: $\cos > 0.85$, $\cos \in (0.5, 0.85]$, $\cos \in (0.15, 0.5]$, $\cos \leq 0.15$.

| N\Solver | $\lceil$WC$\rceil$ | $\lceil$GP$\rceil$ | $\lceil$LP$\rceil$ | $\lceil$SN$\rceil$ | $\lceil$LS$\rceil$ | $\lceil$MM:B$\rceil$ | $\lceil$MM:Bv2$\rceil$ | $\lceil$MM$\rceil$ | $\lceil$MM:R$\rceil$ | $\lceil$DOT$\rceil$ | True |
|---|---|---|---|---|---|---|---|---|---|---|---|
| 1 | ≫10 | 7.69 | 7.80 | 55.50 | 2.16 | 1.52 | 3.48 | ≫10 | 0.04 | 0.73 | 0.00 |
| 16 | ≫10 | 2.12 | 2.33 | 28.93 | 3.79 | 2.10 | ≫10 | ≫10 | 0.16 | 1.43 | 0.00 |

Table 7: $\mathcal{L}^2$↓ metric values of the OT gradient estimated by OT solvers on our **CELEBA IMAGES** benchmark pairs $(\mathbb{P}, \mathbb{Q})$, dimension $D = 12288$. Colors indicate the metric value: $\mathcal{L}^2 < 0.25$, $\mathcal{L}^2 \in [0.25, 0.65)$, $\mathcal{L}^2 \in [0.65, 1.2)$, $\mathcal{L}^2 \geq 1.2$.

| N\Solver | $\lceil$WC$\rceil$ | $\lceil$GP$\rceil$ | $\lceil$LP$\rceil$ | $\lceil$SN$\rceil$ | $\lceil$LS$\rceil$ | $\lceil$MM:B$\rceil$ | $\lceil$MM:Bv2$\rceil$ | $\lceil$MM$\rceil$ | $\lceil$MM:R$\rceil$ | $\lceil$DOT$\rceil$ | True |
|---|---|---|---|---|---|---|---|---|---|---|---|
| 1 | ≫100 | 212.07 | 211.62 | 178.77 | 33.22 | 15.17 | 86.06 | ≫100 | 37.92 | 66.03 | 58.24 |
| 16 | ≫100 | 64.38 | 66.01 | 78.77 | 3.15 | 2.60 | 51.05 | ≫100 | 41.24 | 53.54 | 29.78 |

Table 8: $\mathbb{W}_1$ metric values estimated by OT solvers on our **CELEBA IMAGES** benchmark pairs $(\mathbb{P}, \mathbb{Q})$, dimension $D = 12288$. Colors indicate the value of the relative deviation of $\widehat{\mathbb{W}}_1$ from $\mathbb{W}_1$, i.e., dev $\stackrel{def}{=} 100\% \cdot \frac{|\mathbb{W}_1 - \widehat{\mathbb{W}}_1|}{\mathbb{W}_1}$: dev $< 15\%$, dev $\in [15, 30\%)$, dev $\in [30, 50)\%$, dev $\geq 50\%$

| D\N | 4 | 16 | 64 | 256 |
|---|---|---|---|---|
| 2 | 0.07 | 0.07 | 0.07 | 0.01 |
| 4 | 0.27 | 0.09 | 0.09 | 0.04 |
| 8 | 0.34 | 0.21 | 0.12 | 0.1 |
| 16 | 0.33 | 0.24 | 0.15 | 0.16 |
| 32 | 0.4 | 0.2 | 0.18 | 0.17 |
| 64 | 0.39 | 0.2 | 0.13 | 0.14 |
| 128 | 0.36 | 0.21 | 0.14 | 0.11 |

(a) $\lfloor$**WC**$\rceil$

| D\N | 4 | 16 | 64 | 256 |
|---|---|---|---|---|
| 2 | 0.84 | 0.68 | 0.35 | 0.19 |
| 4 | 0.95 | 0.86 | 0.67 | 0.37 |
| 8 | 0.96 | 0.92 | 0.82 | 0.59 |
| 16 | 0.96 | 0.91 | 0.78 | 0.61 |
| 32 | 0.95 | 0.9 | 0.73 | 0.56 |
| 64 | 0.92 | 0.75 | 0.61 | 0.53 |
| 128 | 0.92 | 0.8 | 0.64 | 0.56 |

(b) $\lfloor$**GP**$\rceil$

| D\N | 4 | 16 | 64 | 256 |
|---|---|---|---|---|
| 2 | 0.82 | 0.81 | 0.68 | 0.49 |
| 4 | 0.95 | 0.9 | 0.82 | 0.66 |
| 8 | 0.96 | 0.92 | 0.83 | 0.68 |
| 16 | 0.96 | 0.92 | 0.78 | 0.6 |
| 32 | 0.95 | 0.9 | 0.73 | 0.55 |
| 64 | 0.92 | 0.76 | 0.6 | 0.53 |
| 128 | 0.93 | 0.75 | 0.62 | 0.56 |

(c) $\lfloor$**LP**$\rceil$

| D\N | 4 | 16 | 64 | 256 |
|---|---|---|---|---|
| 2 | 0.21 | 0.16 | 0.07 | 0.04 |
| 4 | 0.36 | 0.24 | 0.15 | 0.1 |
| 8 | 0.35 | 0.26 | 0.19 | 0.13 |
| 16 | 0.35 | 0.25 | 0.16 | 0.18 |
| 32 | 0.42 | 0.22 | 0.14 | 0.14 |
| 64 | 0.43 | 0.22 | 0.12 | 0.11 |
| 128 | 0.37 | 0.23 | 0.14 | 0.09 |

(d) $\lfloor$**SN**$\rceil$

| D\N | 4 | 16 | 64 | 256 |
|---|---|---|---|---|
| 2 | 0.77 | 0.69 | 0.66 | 0.55 |
| 4 | 0.91 | 0.73 | 0.5 | 0.31 |
| 8 | 0.89 | 0.75 | 0.47 | 0.26 |
| 16 | 0.97 | 0.85 | 0.57 | 0.4 |
| 32 | 0.98 | 0.94 | 0.74 | 0.53 |
| 64 | 0.96 | 0.95 | 0.83 | 0.66 |
| 128 | 0.93 | 0.92 | 0.64 | 0.61 |

(e) $\lfloor$**SO**$\rceil$

| D\N | 4 | 16 | 64 | 256 |
|---|---|---|---|---|
| 2 | 0.56 | 0.44 | 0.11 | 0.05 |
| 4 | 0.7 | 0.38 | 0.12 | 0.0 |
| 8 | 0.36 | 0.19 | -0.07 | -0.13 |
| 16 | -0.16 | -0.23 | -0.24 | -0.24 |
| 32 | -0.36 | -0.42 | -0.42 | -0.38 |
| 64 | -0.54 | -0.48 | -0.47 | -0.45 |
| 128 | -0.53 | -0.55 | -0.53 | -0.51 |

(f) $\lfloor$**LS**$\rceil$

| D\N | 4 | 16 | 64 | 256 |
|---|---|---|---|---|
| 2 | 0.86 | 0.83 | 0.65 | 0.42 |
| 4 | 0.91 | 0.87 | 0.75 | 0.52 |
| 8 | 0.75 | 0.7 | 0.52 | 0.26 |
| 16 | 0.23 | 0.09 | -0.03 | -0.18 |
| 32 | -0.27 | -0.37 | -0.4 | -0.38 |
| 64 | -0.53 | -0.49 | -0.48 | -0.47 |
| 128 | -0.57 | -0.57 | -0.55 | -0.54 |

(g) $\lfloor$**MM:B**$\rceil$

| D\N | 4 | 16 | 64 | 256 |
|---|---|---|---|---|
| 2 | 0.85 | 0.8 | 0.63 | 0.48 |
| 4 | 0.96 | 0.9 | 0.78 | 0.55 |
| 8 | 0.94 | 0.87 | 0.69 | 0.5 |
| 16 | 0.83 | 0.69 | 0.58 | 0.46 |
| 32 | 0.56 | 0.61 | 0.52 | 0.42 |
| 64 | 0.54 | 0.6 | 0.51 | 0.46 |
| 128 | 0.5 | 0.47 | 0.47 | 0.45 |

(h) $\lfloor$**MM:Bv2**$\rceil$

| D\N | 4 | 16 | 64 | 256 |
|---|---|---|---|---|
| 2 | 0.8 | 0.75 | 0.58 | 0.44 |
| 4 | 0.91 | 0.74 | 0.58 | 0.56 |
| 8 | 0.95 | 0.74 | 0.52 | 0.54 |
| 16 | 0.94 | 0.83 | 0.62 | 0.44 |
| 32 | 0.89 | 0.78 | 0.58 | 0.36 |
| 64 | 0.66 | 0.26 | 0.46 | 0.46 |
| 128 | 0.49 | 0.61 | 0.24 | 0.53 |

(i) $\lfloor$**MM**$\rceil$

| D\N | 4 | 16 | 64 | 256 |
|---|---|---|---|---|
| 2 | 0.68 | 0.62 | 0.46 | 0.29 |
| 4 | 0.92 | 0.8 | 0.64 | 0.45 |
| 8 | 0.97 | 0.9 | 0.74 | 0.56 |
| 16 | 0.96 | 0.92 | 0.73 | 0.53 |
| 32 | 0.94 | 0.87 | 0.73 | 0.47 |
| 64 | 0.68 | 0.54 | 0.64 | 0.5 |
| 128 | 0.51 | 0.51 | 0.39 | 0.31 |

(j) $\lfloor$**MM:R**$\rceil$

| D\N | 4 | 16 | 64 | 256 |
|---|---|---|---|---|
| 2 | 0.86 | 0.84 | 0.72 | 0.52 |
| 4 | 0.84 | 0.77 | 0.66 | 0.47 |
| 8 | 0.56 | 0.47 | 0.33 | 0.2 |
| 16 | 0.18 | 0.12 | 0.06 | 0.0 |
| 32 | -0.09 | -0.13 | -0.15 | -0.15 |
| 64 | -0.28 | -0.26 | -0.26 | -0.26 |
| 128 | -0.35 | -0.36 | -0.35 | -0.34 |

(k) $\lfloor$**DOT**$\rceil$

| D\N | 4 | 16 | 64 | 256 |
|---|---|---|---|---|
| 2 | 1.00 | 1.00 | 1.00 | 1.00 |
| 4 | 1.00 | 1.00 | 1.00 | 1.00 |
| 8 | 1.00 | 1.00 | 1.00 | 1.00 |
| 16 | 1.00 | 1.00 | 1.00 | 1.00 |
| 32 | 1.00 | 1.00 | 1.00 | 1.00 |
| 64 | 1.00 | 1.00 | 1.00 | 1.00 |
| 128 | 1.00 | 1.00 | 1.00 | 1.00 |

(l) Ground truth

Table 9: Cos↑ metric values of the OT gradient estimated by OT solvers on our **HIGH-DIMENSIONAL** benchmark pairs $(\mathbb{P}, \mathbb{Q})$. Colors indicate the metric value: $\cos > 0.85$, $\cos \in (0.5, 0.85]$, $\cos \in (0.15, 0.5]$, $\cos \leq 0.15$.

| D\N | 4 | 16 | 64 | 256 |
|---|---|---|---|---|
| 2 | 0.85 | 0.89 | 0.85 | 0.82 |
| 4 | 1.14 | 1.09 | 1.15 | 1.08 |
| 8 | 2.02 | 2.44 | 1.98 | 1.59 |
| 16 | 6.9 | 9.5 | 3.42 | 1.77 |
| 32 | 12.35 | 10.05 | 4.42 | 2.45 |
| 64 | 26.25 | 22.06 | 12.8 | 5.35 |
| 128 | 556.6 | 568.87 | 346.4 | 175.97 |

(a) $\lfloor \mathbf{WC} \rceil$

| D\N | 4 | 16 | 64 | 256 |
|---|---|---|---|---|
| 2 | 0.09 | 0.45 | 1.08 | 1.4 |
| 4 | 0.09 | 0.24 | 0.6 | 1.21 |
| 8 | 0.09 | 0.18 | 0.41 | 0.82 |
| 16 | 0.09 | 0.2 | 0.46 | 0.83 |
| 32 | 0.11 | 0.24 | 0.58 | 0.92 |
| 64 | 0.18 | 0.53 | 0.82 | 0.96 |
| 128 | 0.17 | 0.44 | 0.74 | 0.89 |

(b) $\lfloor \mathbf{GP} \rceil$

| D\N | 4 | 16 | 64 | 256 |
|---|---|---|---|---|
| 2 | 0.17 | 0.15 | 0.34 | 0.63 |
| 4 | 0.09 | 0.17 | 0.3 | 0.54 |
| 8 | 0.1 | 0.17 | 0.35 | 0.62 |
| 16 | 0.09 | 0.19 | 0.46 | 0.79 |
| 32 | 0.12 | 0.23 | 0.56 | 0.92 |
| 64 | 0.17 | 0.52 | 0.83 | 0.98 |
| 128 | 0.16 | 0.52 | 0.77 | 0.9 |

(c) $\lfloor \mathbf{LP} \rceil$

| D\N | 4 | 16 | 64 | 256 |
|---|---|---|---|---|
| 2 | 0.74 | 0.84 | 0.82 | 0.79 |
| 4 | 0.85 | 0.93 | 1.02 | 0.99 |
| 8 | 1.08 | 1.16 | 1.11 | 1.1 |
| 16 | 1.23 | 1.35 | 1.32 | 1.12 |
| 32 | 1.16 | 1.54 | 1.54 | 1.27 |
| 64 | 1.13 | 1.56 | 1.61 | 1.39 |
| 128 | 1.25 | 1.53 | 1.7 | 1.57 |

(d) $\lfloor \mathbf{SN} \rceil$

| D\N | 4 | 16 | 64 | 256 |
|---|---|---|---|---|
| 2 | 0.21 | 0.32 | 0.45 | 0.56 |
| 4 | 0.13 | 0.44 | 0.7 | 0.87 |
| 8 | 0.18 | 0.43 | 0.83 | 1.26 |
| 16 | 0.06 | 0.28 | 0.77 | 1.09 |
| 32 | 0.03 | 0.12 | 0.5 | 0.87 |
| 64 | 0.08 | 0.11 | 0.31 | 0.68 |
| 128 | 0.13 | 0.15 | 0.73 | 0.77 |

(e) $\lfloor \mathbf{SO} \rceil$

| D\N | 4 | 16 | 64 | 256 |
|---|---|---|---|---|
| 2 | 0.51 | 0.69 | 0.81 | 0.79 |
| 4 | 0.58 | 0.82 | 0.97 | 1.0 |
| 8 | 0.83 | 0.98 | 1.12 | 1.15 |
| 16 | 1.28 | 1.37 | 1.39 | 1.41 |
| 32 | 1.61 | 1.68 | 1.69 | 1.64 |
| 64 | 1.93 | 1.87 | 1.87 | 1.86 |
| 128 | 2.1 | 2.12 | 2.1 | 2.07 |

(f) $\lfloor \mathbf{LS} \rceil$

| D\N | 4 | 16 | 64 | 256 |
|---|---|---|---|---|
| 2 | 0.07 | 0.14 | 0.4 | 0.62 |
| 4 | 0.15 | 0.23 | 0.44 | 0.71 |
| 8 | 0.4 | 0.52 | 0.72 | 0.94 |
| 16 | 0.96 | 1.05 | 1.12 | 1.19 |
| 32 | 1.4 | 1.48 | 1.51 | 1.51 |
| 64 | 1.81 | 1.77 | 1.78 | 1.77 |
| 128 | 2.0 | 2.02 | 2.0 | 1.98 |

(g) $\lfloor \mathbf{MM:B} \rceil$

| D\N | 4 | 16 | 64 | 256 |
|---|---|---|---|---|
| 2 | 0.08 | 0.14 | 0.51 | 0.82 |
| 4 | 0.06 | 0.17 | 0.44 | 1.18 |
| 8 | 0.28 | 0.47 | 1.01 | 2.56 |
| 16 | 2.18 | 2.24 | 2.84 | 4.78 |
| 32 | 11.0 | 6.05 | 6.45 | 9.52 |
| 64 | 23.54 | 13.24 | 12.27 | 14.97 |
| 128 | 95.33 | 56.65 | 19.0 | 25.47 |

(h) $\lfloor \mathbf{MM:Bv2} \rceil$

| D\N | 4 | 16 | 64 | 256 |
|---|---|---|---|---|
| 2 | 1.29 | 1.94 | 1.76 | 1.71 |
| 4 | 1.63 | 1.78 | 9.27 | 5.15 |
| 8 | 1.28 | 1.12 | 17.12 | 10.74 |
| 16 | 0.19 | 0.64 | 1.92 | 7.97 |
| 32 | 0.27 | 0.7 | 2.54 | 5.37 |
| 64 | 1.18 | 26.94 | 3.07 | 1.75 |
| 128 | 2.16 | 0.91 | 4.7 | 0.73 |

(i) $\lfloor \mathbf{MM} \rceil$

| D\N | 4 | 16 | 64 | 256 |
|---|---|---|---|---|
| 2 | 0.53 | 0.61 | 0.9 | 1.21 |
| 4 | 0.14 | 0.38 | 0.71 | 1.08 |
| 8 | 0.06 | 0.19 | 0.52 | 0.9 |
| 16 | 0.07 | 0.16 | 0.53 | 0.93 |
| 32 | 0.13 | 0.25 | 0.53 | 1.06 |
| 64 | 0.63 | 0.92 | 0.72 | 1.0 |
| 128 | 0.98 | 0.99 | 1.22 | 1.38 |

(j) $\lfloor \mathbf{MM:R} \rceil$

| D\N | 4 | 16 | 64 | 256 |
|---|---|---|---|---|
| 2 | 0.16 | 0.22 | 0.38 | 0.69 |
| 4 | 0.3 | 0.44 | 0.67 | 1.02 |
| 8 | 0.88 | 1.05 | 1.31 | 1.59 |
| 16 | 1.67 | 1.77 | 1.9 | 1.99 |
| 32 | 2.17 | 2.27 | 2.3 | 2.3 |
| 64 | 2.56 | 2.53 | 2.53 | 2.53 |
| 128 | 2.7 | 2.72 | 2.7 | 2.68 |

(k) $\lfloor \mathbf{DOT} \rceil$

| D\N | 4 | 16 | 64 | 256 |
|---|---|---|---|---|
| 2 | 0.00 | 0.00 | 0.00 | 0.00 |
| 4 | 0.00 | 0.00 | 0.00 | 0.00 |
| 8 | 0.00 | 0.00 | 0.00 | 0.00 |
| 16 | 0.00 | 0.00 | 0.00 | 0.00 |
| 32 | 0.00 | 0.00 | 0.00 | 0.00 |
| 64 | 0.00 | 0.00 | 0.00 | 0.00 |
| 128 | 0.00 | 0.00 | 0.00 | 0.00 |

(l) Ground truth

Table 10: $\mathcal{L}^2 \downarrow$ metric values of the OT gradient estimated by OT solvers on our **HIGH-DIMENSIONAL** benchmark pairs $(\mathbb{P}, \mathbb{Q})$. Colors indicate the metric value: $\mathcal{L}^2 < 0.25$, $\mathcal{L}^2 \in [0.25, 0.65)$, $\mathcal{L}^2 \in [0.65, 1.2)$, $\mathcal{L}^2 \geq 1.2$.

| D\N | 4 | 16 | 64 | 256 |
|---|---|---|---|---|
| 2 | 0.03 | 0.04 | -0.01 | -0.01 |
| 4 | 0.41 | 0.19 | 0.06 | 0.02 |
| 8 | 1.11 | 0.6 | 0.21 | 0.09 |
| 16 | 1.72 | 1.59 | 0.52 | 0.29 |
| 32 | 2.85 | 1.0 | 0.52 | 0.3 |
| 64 | 3.31 | 1.31 | 0.7 | 0.36 |
| 128 | 9.44 | 5.91 | 2.8 | 1.09 |

(a) $\lfloor$**WC**$\rceil$

| D\N | 4 | 16 | 64 | 256 |
|---|---|---|---|---|
| 2 | 0.86 | 0.42 | 0.13 | 0.04 |
| 4 | 1.45 | 1.06 | 0.58 | 0.22 |
| 8 | 2.0 | 1.7 | 1.16 | 0.71 |
| 16 | 2.07 | 1.81 | 1.36 | 0.91 |
| 32 | 1.98 | 1.71 | 1.19 | 0.87 |
| 64 | 1.38 | 1.09 | 0.81 | 0.67 |
| 128 | 1.14 | 0.92 | 0.71 | 0.6 |

(b) $\lfloor$**GP**$\rceil$

| D\N | 4 | 16 | 64 | 256 |
|---|---|---|---|---|
| 2 | 0.89 | 0.48 | 0.22 | 0.1 |
| 4 | 1.48 | 1.06 | 0.69 | 0.39 |
| 8 | 1.98 | 1.66 | 1.18 | 0.78 |
| 16 | 2.12 | 1.81 | 1.35 | 0.91 |
| 32 | 1.92 | 1.61 | 1.18 | 0.8 |
| 64 | 1.46 | 1.15 | 0.83 | 0.69 |
| 128 | 1.14 | 0.87 | 0.69 | 0.64 |

(c) $\lfloor$**LP**$\rceil$

| D\N | 4 | 16 | 64 | 256 |
|---|---|---|---|---|
| 2 | 0.15 | 0.08 | 0.0 | 0.0 |
| 4 | 0.44 | 0.18 | 0.08 | 0.02 |
| 8 | 0.59 | 0.34 | 0.18 | 0.08 |
| 16 | 0.63 | 0.46 | 0.19 | 0.14 |
| 32 | 0.73 | 0.29 | 0.19 | 0.12 |
| 64 | 0.65 | 0.32 | 0.15 | 0.12 |
| 128 | 0.37 | 0.29 | 0.19 | 0.1 |

(d) $\lfloor$**SN**$\rceil$

| D\N | 4 | 16 | 64 | 256 |
|---|---|---|---|---|
| 2 | 0.76 | 0.28 | 0.09 | 0.04 |
| 4 | 1.27 | 0.56 | 0.26 | 0.09 |
| 8 | 1.68 | 1.04 | 0.54 | 0.29 |
| 16 | 1.89 | 1.46 | 0.85 | 0.55 |
| 32 | 1.78 | 1.52 | 1.06 | 0.73 |
| 64 | 1.43 | 1.37 | 1.08 | 0.8 |
| 128 | 1.05 | 0.98 | 0.69 | 0.65 |

(e) $\lfloor$**SO**$\rceil$

| D\N | 4 | 16 | 64 | 256 |
|---|---|---|---|---|
| 2 | 0.43 | 0.09 | 0.0 | 0.0 |
| 4 | 0.45 | 0.15 | 0.01 | -0.01 |
| 8 | 0.24 | 0.07 | -0.06 | -0.06 |
| 16 | -0.16 | -0.2 | -0.2 | -0.18 |
| 32 | -0.36 | -0.37 | -0.35 | -0.29 |
| 64 | -0.43 | -0.37 | -0.35 | -0.32 |
| 128 | -0.38 | -0.36 | -0.34 | -0.32 |

(f) $\lfloor$**LS**$\rceil$

| D\N | 4 | 16 | 64 | 256 |
|---|---|---|---|---|
| 2 | 0.79 | 0.44 | 0.19 | 0.06 |
| 4 | 1.08 | 0.67 | 0.35 | 0.11 |
| 8 | 0.74 | 0.52 | 0.21 | 0.05 |
| 16 | 0.11 | 0.0 | -0.05 | -0.1 |
| 32 | -0.23 | -0.27 | -0.26 | -0.24 |
| 64 | -0.39 | -0.35 | -0.33 | -0.32 |
| 128 | -0.38 | -0.37 | -0.35 | -0.33 |

(g) $\lfloor$**MM:B**$\rceil$

| D\N | 4 | 16 | 64 | 256 |
|---|---|---|---|---|
| 2 | 0.8 | 0.47 | 0.21 | 0.12 |
| 4 | 1.42 | 1.01 | 0.74 | 0.52 |
| 8 | 2.09 | 1.82 | 1.46 | 1.21 |
| 16 | 2.86 | 2.54 | 2.14 | 1.68 |
| 32 | 4.05 | 3.16 | 2.37 | 2.03 |
| 64 | 4.6 | 3.58 | 2.71 | 2.61 |
| 128 | 7.2 | 3.91 | 2.37 | 2.56 |

(h) $\lfloor$**MM:Bv2**$\rceil$

| D\N | 4 | 16 | 64 | 256 |
|---|---|---|---|---|
| 2 | 0.83 | 0.48 | 0.27 | 0.15 |
| 4 | 1.35 | 1.12 | 1.35 | 0.91 |
| 8 | 1.85 | 1.58 | 2.75 | 2.1 |
| 16 | 1.96 | 1.76 | 1.74 | 1.87 |
| 32 | 1.76 | 1.69 | 1.68 | 1.35 |
| 64 | 1.22 | 1.84 | 1.17 | 0.79 |
| 128 | 0.97 | 0.76 | 0.6 | 0.33 |

(i) $\lfloor$**MM**$\rceil$

| D\N | 4 | 16 | 64 | 256 |
|---|---|---|---|---|
| 2 | 0.79 | 0.47 | 0.28 | 0.16 |
| 4 | 1.37 | 0.99 | 0.66 | 0.43 |
| 8 | 1.89 | 1.54 | 1.28 | 1.04 |
| 16 | 2.06 | 1.84 | 1.8 | 1.55 |
| 32 | 2.26 | 2.01 | 2.22 | 1.8 |
| 64 | 3.32 | 4.35 | 1.51 | 1.02 |
| 128 | 0.6 | 0.73 | 0.56 | 0.39 |

(j) $\lfloor$**MM:R**$\rceil$

| D\N | 4 | 16 | 64 | 256 |
|---|---|---|---|---|
| 2 | 0.82 | 0.47 | 0.24 | 0.15 |
| 4 | 1.43 | 1.07 | 0.8 | 0.64 |
| 8 | 2.42 | 2.19 | 1.92 | 1.79 |
| 16 | 4.13 | 4.04 | 3.98 | 3.93 |
| 32 | 7.26 | 7.23 | 7.23 | 7.26 |
| 64 | 12.01 | 12.05 | 12.07 | 12.09 |
| 128 | 18.88 | 18.9 | 18.92 | 18.93 |

(k) $\lfloor$**DOT**$\rceil$

| D\N | 4 | 16 | 64 | 256 |
|---|---|---|---|---|
| 2 | 0.82 | 0.47 | 0.23 | 0.13 |
| 4 | 1.39 | 1 | 0.71 | 0.49 |
| 8 | 1.88 | 1.59 | 1.26 | 1.01 |
| 16 | 1.94 | 1.74 | 1.55 | 1.4 |
| 32 | 1.82 | 1.67 | 1.54 | 1.41 |
| 64 | 1.41 | 1.36 | 1.32 | 1.28 |
| 128 | 1.14 | 1.07 | 1.04 | 1.04 |

(l) Ground truth

Table 11: $\mathbb{W}_1$ values estimated by OT solvers on our **HIGH-DIMENSIONAL** benchmark pairs $(\mathbb{P}, \mathbb{Q})$. Colors indicate the value of the relative deviation of $\widehat{\mathbb{W}}_1$ from $\mathbb{W}_1$, i.e., dev $\overset{def}{=} 100\% \cdot \frac{|\mathbb{W}_1 - \widehat{\mathbb{W}}_1|}{\mathbb{W}_1}$: dev $< 15\%$, dev $\in [15, 30\%)$, dev $\in [30, 50)\%$, dev $\geq 50\%$.

# D   Transport Rays

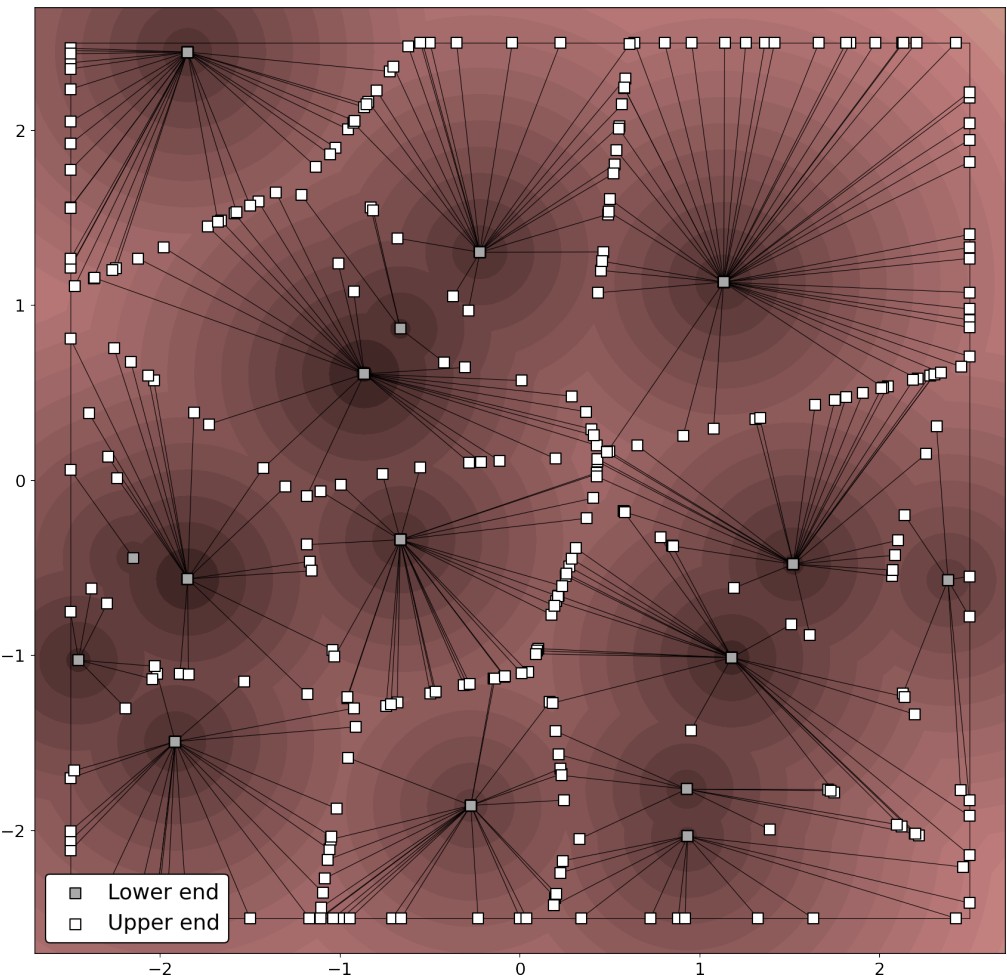

Figure 9: Truncated transport rays of a random MinFunnel in dimension $D = 2$ with $N = 16$.