# OpenReview forum: "Kantorovich Strikes Back! Wasserstein GANs are not Optimal Transport?"
_NeurIPS.cc/2022/Track/Datasets_and_Benchmarks — NeurIPS 2022 Datasets and Benchmarks _

### Official Review · Reviewer_kVP9 · 2022-07-09
**Important setting quantifying W1 approximations**

**Rating:** 7
**Confidence:** 3
**Correctness:** The benchmark and methods are correct…
**Clarity:** The paper is well-written and easy to…

**Strengths:**

+ Approximating W1 computations is widely used and a difficult
  setting to benchmark because the ground-truth transport
  maps and distances are often not known.
  I am not aware of an established W1 benchmarks and papers
  often have to rely on downstream tasks (such as inception scores)
  to justify an algorithmic improvement to the W1 approximation.
+ This paper presents non-trivial settings where the
  ground-truth transport map is known and uses it to
+ The experimental results are thorough and the paper
  strongly shows that minimax methods solve
  the benchmark tasks in most settings, at least for
  obtaining a gradient that approximates the true gradient.

**Weaknesses:**

+ While the paper proposes a new benchmark for approximating
  the W1, it unfortunately does not present results
  in established GAN settings as the ground-truth maps
  are not known. Thus research that is ultimately focused on
  improving the W1 computations in settings such as GANs
  may be able to use these benchmarks for preliminary
  experiments, but these benchmark tasks may not
  reflect the true difficulties.
  of these methods thus established and powerful
+ It is not clear how "solved" W1 OT is, how much work
  remains in the field, and how many new directions
  this benchmark will enable. In other words, better
  solutions to this benchmark will not directly enable
  new methods (or new GAN results).


**Additional Feedback:**

n/a

**Documentation:**

The code is well-documented with enough detail to show how others can build new algorithms on top of it. I have one suggestion: it would be much easier to use if the main package was hosted on pip so that new methods could be developed more easily in a separate repository.

**Relation To Prior Work:**

This paper properly discusses and connects together all of the relevant work

**Summary And Contributions:**

This paper proposes a benchmark for methods computing the
Wasserstein-1 distance. Section 1 summarizes background
information on computing W1, often with
the dual in eq (4) and (5), and how the W1 is used
in GAN training.
Section 2 summarizes methods estimating the dual
potentials and transport maps.
Section 3 describes the benchmark distributions,
and Section 4 shows the results of evaluating the
methods on the results, which are quantified in
Section D of the appendix.

---

> ### Author Response · Authors · 2022-08-18
> **Answer to Reviewer kVP9**
>
> Thank you for your valuable feedback. Please find above (in our reply to all the Reviewers) the answers to your comments common with other reviews. Please find below our answers to your questions that do not overlap with those of other Reviewers.
>
> **(1a) While the paper proposes a new benchmark for approximating the W1, it unfortunately does not present results in established GAN settings as the ground-truth maps are not known. Thus research that is ultimately focused on improving the W1 computations in settings such as GANs may be able to use these benchmarks for preliminary experiments, but these benchmark tasks may not reflect the true difficulties. of these methods thus established and powerful.**
>
> **(1b) It is not clear how "solved" W1 OT is, how much work remains in the field, and how many new directions this benchmark will enable. In other words, better solutions to this benchmark will not directly enable new methods (or new GAN results)**
>
> We agree with you comment that studying how the OT solvers work in WGANs would be a nice addition to the paper. We think that this question is beyond the scope of our OT benchmark and leave it open for future studies. We added a discussion about this question to Section 4 of the revised version of the paper (limitations/future work).
>
> Please note that the main goal of our benchmark is not to understand how to improve OT solvers for WGANs. We aim to evaluate how well these solvers compute OT. This is different and is not necessarily connected to the performance of the solvers in GANs, e.g., see the conclusions of the related Wasserstein-2 benchmark [23]. The exposition of our paper indeed considers WGANs, but this is simply because most OT solvers come from the WGAN literature.
>
> Due to the high popularity of the Wasserstein-1 loss and its success in GANs (Karras et al., 2020), in ML community it is commonly assumed that the underlying solvers successfully compute OT.
> This opinion is actively propagated in many research papers, educational courses, scientific or educational YouTube videos, community blog posts, etc. Despite this, **there is no high-dimensional dataset for the $\mathbb{W}_{1}$ evaluation to support/reject this claim** (see lines 80-87). Hence **our method fills this important gap**, thereby allowing quantitative evaluation of future related research. As a result, this development allows us to verify the common opinion that WGAN solvers successfully compute OT.
>
>
>
> **Concluding remarks**. Please respond to our post to let us know if the clarifications above suitably address your concerns about our work. We are happy to address any remaining points during the discussion phase.

---

> > ### Comment · Reviewer_kVP9 · 2022-08-18
> > **Response**
> >
> > Thank you for the response and additional information! I've read through all of the other reviewing information and still think the paper should be accepted

---

### Official Review · Reviewer_7yDg · 2022-07-20
**An interesting benchmark of dual OT solver for W1 distance**

**Rating:** 7
**Confidence:** 3
**Clarity:** This paper is well written and easy t…

**Strengths:**

- This papers proposes a method to build **known** OT maps using 1-Lipschitz MinFunnet functions. This choice is clearly justified as these functions are universal approximator of 1-Lipschitz functions (Prop.2). Having known OT maps allows to faithfully compare the OT solvers
- They carefully build transport ray of these functions.
- The paper is well written and easy to follow.
- The authors tackle an interesting problem and having more comparison like this one is crucial

**Weaknesses:**

- I regret that the results of the benchmarks are only available in the Appendices. I would recommend the authors to include some of them in the main paper since those are the main results of the paper.
- The restriction to 1-Lipschitz *MinFunnet* functions seems to be a main limitation of this work.
- It seems that in the experiments only one random start is considered. Is there any reasons why the authors did not perform multiple runs? This seems to impede to assess the methods stability and robustness with regard to the random start and the parameters $a_n$ and $b_n$ in the *funnel*.

**Additional Feedback:**

- I would suggest the authors reshaping a bit the paper to include some of the metrics from the benchmarks in the main paper to better support the claims made when discussing the results in Sec.4.
- If there any reason why the authors did not include any error bars on the metrics they computed?

**Correctness:**

- The benchmarks pairs are well designed and well justified.
- Not including error bars in the experiments prevents from quantifying the methods stability and robustness to the random start. This should be part of the benchmark.

**Documentation:**

The github repo has a `README.md` file explaining how to reproduce the results from the notebooks.

**Ethics:**

The datasets used in this paper are well known and are free to use.

**Relation To Prior Work:**

To the best of my knowledge this paper clearly discusses and states how it differs from prior works.

**Summary And Contributions:**

This paper proposes a benchmark for computing the Wasserstein-1 distance. The authors first propose to use 1-Lipschitz functions to build ray monotone transport plans and obtain known OT maps. These ground truth maps are then used to benchmark dual OT solvers used in particular in the Wasserstein GAN framework.

---

> ### Author Response · Authors · 2022-08-18
> **Answer to Reviewer 7yDg**
>
> Thank you for your valuable feedback. Please find above (in our reply to all the Reviewers) the answers to your comments common with other reviews. Please find below our answers to your questions that do not overlap with those of other Reviewers.
>
> **(1) The restriction to 1-Lipschitz MinFunnel functions seems to be a main limitation of this work**
>
> We use MinFunnels because their transport rays admit closed form. However, our main idea to create benchmark pairs by using $1$-Lipschitz functions $u$ is rather generic and can be applied to any $1$-Lipschitz function as soon as we know how to compute $u$'s transport rays. Developing methods to compute transport rays for more generic $1$-Lipschitz functions $u$ is a promising direction for future studies which could help to further advance the benchmark.
>
>
> **(2) It seems that in the experiments only one random start is considered. Is there any reasons why the authors did not perform multiple runs? This seems to impede to assess the methods stability and robustness with regard to the random start and the parameters**
>
> Initially, we did not include results for multiple random restarts because, in our view, the provided evaluation is enough to understand the main trends of WGAN OT solvers's performance. For most solvers their performance smoothly drops with the increase of dimension $D$ and number $N$ of funnels. This can be seen from the colored Tables in Appendix D.
>
> To address your concern, we conducted the two following experiments with varying random seeds on our high-dimensional benchmark pairs with parameters $(N,D)\in \lbrace (16,32), (16,64), (64,32), (64,64)\rbrace$:
>
> **(a) Varying seed for the init of pairs (randomness in $a_{n},b_{n}$).** We trained WGAN solvers with a **single** random seed on our high-dimensional benchmark re-initialized with **five** different random seeds pairs.
>
> **(b) Varying seed for WGAN solvers (randomness in nets init, stochasticity in training).** We trained WGAN solvers with **five** random seeds on a **single** random initialization of high-dimensional benchmark pair.
>
>
> The results of $\cos$ metrics ($\mu\pm\sigma$) are shown in the supplementary material (*answers/7yDg\_randomness* folder). In both cases, for the solvers the standard deviation is minor ($\sim 0.01$) with the exception of $\lfloor \mbox{MM}\rceil$, $\lfloor \mbox{MM:R}\rceil$ for which it is higher ($\sim 0.07$) due to instabilities of maximin training. Note we have already highlighted this in Section 4 of the initial submission. Thus, in the main Tables (Appendix D), we report the results only of a single start to keep them clean.
>
> **Concluding remarks**. Please respond to our post to let us know if the clarifications above suitably address your concerns about our work. We are happy to address any remaining points during the discussion phase.

---

> > ### Comment · Reviewer_7yDg · 2022-08-24
> > **Thank you for your answers**
> >
> > I want to thank the authors for their clarifications and for running these additional experiments.
> > I think the new results provided by the authors with different random seeds and on other *real-life* dataset (CIFAR) better show the robustness and stability of the considered methods.
> >
> > I would like to increase my score to 7 and so vote acceptance.

---

### Official Review · Reviewer_Givm · 2022-07-24
**A generic benchmark to evaluate dual OT solvers for the Wasserstein-1 distance.**

**Rating:** 7
**Confidence:** 2
**Clarity:** The paper is clearly written and well…

**Strengths:**

1. This paper proposed a benchmark to evaluate the methods of computing the Wasserstein-1 distance. The problem is interesting to the community.

2. This paper is well-written and technically sound. The method uses 1-Lipschitz functions to construct pairs of continuous distributions, which is well designed.

3. This paper thoroughly evaluates popular WGAN dual form solvers in high-dimensional spaces using these benchmark pairs.

**Weaknesses:**

1. The title of this paper is ambiguous and may lead to inappropriate reviewers.

2. The theoretical analysis and the intuition of the proposed method is weak. It is unclear why the proposed method works well than previous methods.

3. Evaluating the Wasserstein-1 distance does not directly validate the superiority of the methods on specific tasks, which may need more explanations.

**Additional Feedback:**

NA

**Correctness:**

The evaluation methods and experiment design are generally appropriate and performed correctly.

**Documentation:**

The Github repo is provided with a detailed explanation for reproducibility.

**Relation To Prior Work:**

See Weaknesses.

**Summary And Contributions:**

This paper proposes a benchmark to evaluate the methods of computing the Wasserstein-1 distance. The authors construct 1-Lipschitz functions and use them to build ray monotone transport plans, which yield pairs of continuous benchmark distributions in high-dimensional spaces. Some WGAN dual form solvers are evaluated using these benchmark pairs.

---

> ### Author Response · Authors · 2022-08-18
> **Answer to Reviewer Givm**
>
> Thank you for your valuable feedback. Please find above (in our reply to all the Reviewers) the answers to your comments common with other reviews. Please find below our answers to your questions that do not overlap with those of other Reviewers.
>
> **(1) The title of this paper is ambiguous and may lead to inappropriate reviewers**
>
> In the GANs' training, the theoretical aspects of the underlying loss are typically sacrificed in favor of regularizations, normalizations, approximations improving the stability of training. Thought the current title, we would like to emphasize that **what remains** in the WGAN solvers after employing these tricks in some cases barely represents the initial OT loss and its gradient which are theoretically justified and motivated.
>
>
>
> **(2) The theoretical analysis and the intuition of the proposed method is weak. It is unclear why the proposed method works well than previous methods.**
>
> As we write in lines 35-38, 84-87, prior approaches  to evaluate WGAN [31,48,40] solvers focused exclusively on testing how the solvers evaluate $\mathbb{W}_{1}(\mathbb{P},\mathbb{Q})$. They used discrete distributions $\mathbb{P},\mathbb{Q}$ with the ground truth OT plan/cost computed by a discrete OT solver. Their methodology is not applicable to testing how well the solvers compute the OT gradient $\nabla f^{*}$ which is needed for WGAN training because this gradient is {ill-defined} in the discrete case (lines 84-87).
>
> Our generic methodology constructs **continuous** benchmark distributions $\mathbb{P},\mathbb{Q}$ with the known OT cost and the unique and **well-defined** OT gradient. Therefore, *unlike the prior approaches for benchmarking $\mathbb{W}_{1}$ solvers, our developed methodology enables the quantitative analysis of the OT gradient computed by OT solvers.*
>
> Could you please clarify, which part of the proposed theoretical analysis is weak and requires further development?
>
> **(3) Evaluating the Wasserstein-1 distance does not directly validate the superiority of the methods on specific tasks, which may need more explanations.**
>
> Please note that the goal of our benchmark is not to understand which solver is superior as the loss for WGANs, but to understand **how well these solvers compute OT**. Due to high popularity of WGANs and their success in generative modeling, in ML community it is commonly assumed that the underlying solvers **successfully compute OT**.  This opinion is actively propagated through many research papers, educational courses, scientific or educational YouTube videos, community blog posts, etc. As our benchmark suggests, **this opinion is not entirely correct**.
>
> We agree with your comment that studying how the OT solvers work in WGANs would be a nice addition to the paper. We think that this question is beyond the scope of our OT benchmark and leave it open for future studies. We added a discussion about this question to Section 4 of the revised version of the paper.
>
> **Concluding remarks**. Please respond to our post to let us know if the clarifications above suitably address your concerns about our work. We are happy to address any remaining points during the discussion phase.

---

### Official Review · Reviewer_CJQz · 2022-07-27
**Semi-synthetic W1 benchmark with ground truth**

**Rating:** 7
**Confidence:** 4

**Strengths:**

- Discusses good overview of W1 methods.
- Proves theoretical results about how to construct maps that are optimal w.r.t. W1.
- Proposes novel way to construct ground-truth (semi-)synthetic benchmarks for evaluating Wasserstein-1 dual solvers.
- Provides code and datasets for benchmark datasets and algorithms.
- Evaluates the gradient of the W1 w.r.t. the parameters, which is actually most important for most generative methods.

**Weaknesses:**

- Only one real-world dataset (celebA) is considered. And the synthetic datasets are quite simple (i.e., truncated Gaussians). It seems including more real-world datasets (even MNIST or CIFAR10) would be useful or using interesting real-world tabular data for smaller dimensions (e.g., even something like iris).

- (This limitation is mentioned in the text but does seem to be the main limitation) It seems the benchmark only considers maps where the samples are grouped more closely together (or the reverse). Maps that expand parts of the space or where some expand and some contract would be better. It is unclear whether the benchmark maps properly represent real-world OT maps.

- (Minor but nonetheless important for final paper) All result tables are in the appendix. And the figures are in odd places with nonstandard captions. At least some summary table of the results and your recommendations for suggested methods based on context would be important to include. What methods would you recommend and why?  The answer may be a combination of ease-of-use, convergence behavior, and overall performance.


**Additional Feedback:**

See clarity above.


**Clarity:**

Overall, I found the paper was clearly written but could use some improvements in a few areas.

- Why do you truncate to the hypercube? The reason stated is "For convenience" but I don't understand why it's convenient. Why not just sample from uniform distribution?  Why truncated Gaussian?
- I did not understand why you could choose any point along the ray (lines 194-195).  Does this mean you could choose a different point along the ray for every $x$?  For example suppose two unique points both lie along the same ray, could you choose two different points to map them to (e.g., that switches their order)?  Maybe I'm missing something here. (It seems you choose a smooth, deterministic way in Sec 3.3 Eq 12.)
- The discussion section is fairly hard to parse without a legend for all the methods.  Maybe a small inline table summarizing each method (e.g., "GP - Gradient Penalty") would be helpful. Otherwise it is very hard to remember all the abbreviations.

*Minor Clarity*
- An early illustration in 1D of L1 functions with "rays" and corresponding transport maps could help introduce the main idea.  I found Figure 3 helpful but it might have helped to anchor intuition before diving into the math.
- Figure captions look centered rather than justified and are nonstandard.
- It might be interesting to see overlaid t-SNE plots of Figure 4 that project both samples of P and Q via t-SNE.


**Correctness:**

- Shouldn't MM and MM:R (Eqs 9 and 10) have the min outside the integral?  Or is it meant to find a new $T$ or $H$ for every $x$ or $y$? Maybe I'm missing something about this notation.


**Documentation:**

- Are the pretrained models saved and available in an archival format?


**Ethics:**

No concerns identified.


**Relation To Prior Work:**

Overall, the paper does a good job of placing the contributions w.r.t. related work. However, the related W2 benchmark should probably be mentioned much earlier on and at least some brief explanation of how these are related but different.  The first mention of this was in the experiments---and was actually a question I had while reading the first part.

**Summary And Contributions:**

Motivated by the lack of benchmarks for W1 dual methods (other than perceptual measures such as FID or IS), this paper proposes to create a (semi-)synthetic set of benchmark datasets with known optimal transport plans, maps, and distance.
To do this, the paper first develops theory about maps that are optimal by construction.
Then, the paper proposes concrete methods for constructing the necessary functions and computing the necessary plans, maps and gradients.
Finally, synthetic dataset pairs are generated from truncated Gaussian data and CelebA data at various dimensionalities and used to evaluate and discuss many existing W1 methods.

---

> ### Author Response · Authors · 2022-08-18
> **Answer to Reviewer CJQz**
>
> Thank you for your valuable feedback. Please find above (in our reply to all the Reviewers) the answers to your comments common with other reviews. Please find below our answers to your questions that do not overlap with those of other Reviewers.
>
> **(1)  It seems the benchmark only considers maps where the samples are grouped more closely together (or the reverse). Maps that expand parts of the space or where some expand and some contract would be better. It is unclear whether the benchmark maps properly represent real-world OT maps.**
>
> Our main idea to create benchmark pairs by using $1$-Lipschitz functions $u$ is rather generic. We use MinFunnels because their transport rays admit closed form and MinFunnels are universal approximators of 1-Lipschitz functions (our Proposition 2). Developing algorithms to compute transport rays of more generic $1$-Lipschitz $u$ is a promising research direction which could help to create more realistic benchmarks for $\mathbb{W}_{1}$.
>
> **(2) Why do you truncate to the hypercube? The reason stated is "For convenience" but I don't understand why it's convenient. Why not just sample from uniform distribution? Why truncated Gaussian?**
>
> Thanks a lot for comment, it is misprint. Indeed, we sample from uniform distribution, one can see it in Figure 3(a) and in our repository in src/distributions.py in (lines 17-28). We fixed this misprint in the text of the paper.
>
> **(3) I did not understand why you could choose any point along the ray (lines 194-195). Does this mean you could choose a different point along the ray for every x? For example suppose two unique points both lie along the same ray, could you choose two different points to map them to (e.g., that switches their order)? Maybe I'm missing something here. (It seems you choose a smooth, deterministic way in Sec 3.3 Eq 12.)**
>
> Consider a (finite) ray parameterized as $[0,1]$. The function $u$ along the ray equals $u(x)=x+C$. To be $u$-ray monotone, the transport map $T$ must satisfy $u(x)-u(T(x))=|x-T(x)|$. This holds if and only if $T(x)\in [0, x]$ (please note that **this is not the entire ray**). First, yes, we can choose **different** points $T(x),T(x')$ for different $x\neq x'$: they only need to satisfy $T(x)\in [0,x]$ and $T(x)\in [0,x']$. In particular, it is allowed to switch their order.
>
> In practice, for simplicity, we use the **same** function $T$ (eq. 12) to move points in the similar manner along **every** ray. We note, however, this function may also be chosen to depend on the ray.
>
> **(4) An early illustration in 1D of L1 functions with "rays" and corresponding transport maps could help introduce the main idea. I found Figure 3 helpful but it might have helped to anchor intuition before diving into the math.**
>
> The transport rays of 1-Lipschitz functions $f$ in 1D are sufficiently simple: they are just segments $[x_{0},x_{1}]\subset \mathbb{R}$ where $f$ is linear and $\nabla f(x)=1$ (or $\nabla f(x)=-1$). This is why to better explain the transport rays we focus on illustrating them on 2D surfaces (Figure 1 and Appendix C). We added an early reference to these Figures to Section 3.1.
>
> **(5) It might be interesting to see overlaid t-SNE plots of Figure 4 that project both samples of P and Q via t-SNE.**
>
> Initially, we considered PCA for the dimensionality reduction as it already provided visually distinguishable plots. Following your comment, we also include t-SNE visualizations of benchmarks pairs of Figure 4 to the supplementary material (*answers/CJQz\_tsne\_visualizations* folder). Note, however, PCA plots (in the main text) visually are better.
>
> **(6) Shouldn't MM and MM:R (Eqs 9 and 10) have the min outside the integral?**
>
> Thanks for you comment, your understanding is correct. We fixed this typo in the revision.
>
> **(7) Are the pretrained models saved and available in an archival format?**
>
> We provide the constucted 1-Lipschitz MinFunnel functions $u$ (pytorch **state dict** s with parameters $a_{n},b_{n}$) for all the constructed benchmark pairs in the *.pt* (pytorch model checkpoint) format. Please, follow the instructions in README.md file in the repository to load the benchmark pairs and sample from them.
>
> We do not keep the checkpoints for trained WGAN dual OT solvers on our benchmark pairs. However, one may easily train them by running our provided notebooks *notebooks/test\_$\star$.ipynb* which evaluate OT solvers.
>
> **Concluding remarks**. Please respond to our post to let us know if the clarifications above suitably address your concerns about our work. We are happy to address any remaining points during the discussion phase.

---

### Official Review · Reviewer_PTCv · 2022-07-28
**Is it hard to turn hyperparameters? Can it be applied to real-world images?**

**Rating:** 8
**Confidence:** 2
**Correctness:** The claims and method are correct, ba…
**Clarity:** The paper is well written.

**Strengths:**

The authors provide an elaborate introduction to the Wasserstein-1 and its neural dual OT solvers. Followed by compact math proof about their benchmark pairs. Experiments are also reasonable. It is also a good point of view to consider the gradient of the Wasserstein-1 distance.

**Weaknesses:**

Some minor concerns.

Is it hard to turn hyperparameters for this method? For example, when you compute the High-dimensional benchmark pairs, you choose $b_n \sim \mathcal{N}(0,0.1)$ and p = 8. How do you choose it? How long does it cost for the hyperparameter search?

The dimension of images ,in reality, is higher than $2^7$. Can this tool handle higher dimensions?

If we carefully choose MinFunnel function u, instead of randomly picking, will the performance be better?

What will be the effect of increasing N and D?

Paper mentions "in WGANs, the solvers move the generated distribution (bad images, $\mathbb{Q}$ in our construction) to the real distribution (good images, $\mathbb{P}$)". However, $\mathbb{P}$ is synthetic distribution and $\mathbb{Q}$ is computed ground truth 'real image' distribution, in the case of images benchmark. Why do the solvers move $\mathbb{Q}$ to $\mathbb{P}$, instead of the opposite?

Authors mention solvers MM, MM:R takes longer for training, compared with GP, SO, and LP. Is the time gap significant?

**Additional Feedback:**

N/A

**Documentation:**

There are enough details to reproduce the major results of this work. The authors also provide the code.

**Ethics:**

There is no known ethical issue.

**Relation To Prior Work:**

It is clearly discussed how this work differs from previous contributions. It also introduces important background knowledge.

**Summary And Contributions:**

Authors propose a generic methodology to construct benchmark pairs with ground truth OT plan, OT cost, and OT gradient. We can use this tool to evaluate the performance of the neural dual OT solvers approximating the Wasserstein-1 distance or the gradient of Wasserstein-1 distance.

Specifically, the authors employ the 1-Lipschitz MinFunnel functions to compute transport rays and define the ray monotone map. With them, we can define a target distribution $\mathbb{Q}$ and compute OT cost and OT gradient based on the original distribution $\mathbb{P}$

---

> ### Author Response · Authors · 2022-08-18
> **Answer to Reviewer PTCv**
>
> Thank you for your valuable feedback. Please find above (in our reply to all the Reviewers) the answers to your comments common with other reviews. Please find below our answers to your questions that do not overlap with those of other Reviewers.
>
> **(1) Authors mention solvers MM, MM:R takes longer for training, compared with GP, SO, and LP. Is the time gap significant?**
>
> Following you suggestion, we provided convergence plots ($\cos$ metric as a function of the training time $t$) for these solvers, see the supplementary material (*answers/PTCv\_convergence\_plots* folder). Here we consider 4 benchmark pairs: 2 high-dimensional pairs with $(N,D)\in \lbrace (16,32),(16,64)\rbrace$, Celeba and CIFAR-10 pairs with $(N,p)=(1,10)$.
>
> The results show that the time gap between $\lfloor \mbox{GP}\rceil$, $\lfloor \mbox{LP}\rceil$, $\lfloor\mbox{SO}\rceil$ and $\lfloor \mbox{MM}\rceil$, $\lfloor \mbox{MM:R}\rceil$ is **noticeable**. This is expected as the latter two have inner optimization cycle. Interestingly, $\lfloor\mbox{SO}\rceil$ takes slightly longer to converge than $\lfloor \mbox{GP}\rceil$, $\lfloor \mbox{LP}\rceil$ but provides higher $\cos$ values on the high-dimensional benchmark pairs.
>
> **(2) Is it hard to turn hyperparameters for this method? For example, when you compute the High-dimensional benchmark pairs, you choose $b_{n} \sim \mathcal{N}(0,0.1)$ and $p = 8$. How do you choose it? How long does it cost for the hyperparameter search?**
>
> Constructing the benchmark, we aimed to make $\mathbb{P}$ and $\mathbb{Q}$ visually distinguishable. We found that hyperparameter $p$ is the most responsible for this as it affects how much input samples $x\sim\mathbb{P}$ move along the transport rays. Namely, higher $p$ yields farther movement and for $p\rightarrow\infty$ sample in $y=T(x)\sim \mathbb{Q}$ concentrate near the centers $a_{n}$ of funnels.
>
> In our **high-dimensional** benchmark pairs, we chose $p=8$ as this yields visually distinguishable $\mathbb{P},\mathbb{Q}$ in PCA plots (Figure 4). Preparing **CelebA** pairs, we noted that with the increase of the number of (random) funnels $N$ (for a fixed $p$), the samples move less, i.e., the images $T(x)$ get less distorted. We think this is because the transport rays become shorter as there are more intersections of funnels. After some manual tuning (which took rougly **few hours**) of $(N,p)$ and visually assessing and discussing the results, we picked $(N,p)=(1,10)$ and $(16,100)$ which yield both distinguishable samples (Figures 5ab) from $\mathbb{P}, \mathbb{Q}$ and PCA plots (Figures 5cde).
>
> Regarding parameters $a_{n},b_{n}$, there is no specific procedure for their tuning. Parameter $a_{n}$ is simply picked at random from the hypercube $[-1,1]^{D}$, while $b_{n}\sim \mathcal{N}(0,0.1)$. We did not study their selection, as this initialization methodology already provided the *desirable results regardless of a particular random realization for* $a_{n},b_{n}$.
>
> In our repository, we provide a set of  **ready-to-use benchmark pairs**, i.e., (randomly generated) MinFunnels $\lbrace a_{n},b_{n}\rbrace_{n=1}^{N}$ and the code to use them to sample from respective pairs $\mathbb{P},\mathbb{Q}$. In addition, our code could be used to create, view and test new benchmark pairs (with random or manually initalized $a_{n},b_{n}$).
>
>
> **(3) The dimension of images, in reality, is higher than $2^{7}$. Can this tool handle higher dimensions?**
>
> Our developed approach is generic and can handle **any dimension**. In the paper, we construct several benchmark pairs in dimensions up to $D=2^{7}$. Then, we also construct 2 benchmark pairs on the space with $D=12288$ (CelebA images pairs) -- this is much bigger than $2^{7}$. Additionally, as per Reviewers' Ceo9, CJQz request, we enhanced our benchmark with 2 new benchmark pairs in the space of dimension $3072$ (CIFAR-10 benchmark pairs), see our general response.
>
>
>
> **(4) If we carefully choose MinFunnel function $u$, instead of randomly picking, will the performance be better?**
>
> *Could you please clarify, what exactly do you mean by the performance here?*

---

> > ### Author Response · Authors · 2022-08-18
> > **Answer to Reviewer PTCv (part 2)**
> >
> > **(5) What will be the effect of increasing N and D?**
> >
> > This effect for high-dimensional benchmark is represented in Tables 8, 9 and 10 of the revision. With the increase of N and D, the constructed benchmark pairs naturally become more complex and the solvers' error increases.
> >
> > **(6) Paper mentions "in WGANs, the solvers move the generated distribution (bad images, Q in our construction) to the real distribution (good images, P)". However, P is synthetic distribution and Q is computed ground truth 'real image' distribution, in the case of images benchmark. Why do the solvers move Q to P, instead of the opposite?**
> >
> > We think that there might be a slight misunderstanding of our images benchmark pairs. It is not possible to directly represent a WGAN training setup as the true OT gradient from fake (generated) data to real data is unknown. Thus, in our CelebA benchmark pairs, we **simulate** two distributions with a known OT plan -- a good one (*real images*) $\mathbb{P}$ and a bad one (*fake images*) $\mathbb{Q}$. Pair $(\mathbb{Q}\rightarrow \mathbb{P})$ is then used to assess how well the OT solvers recover the OT gradient from bad to good images. This setup is intuitively related to WGAN training but **does not exactly represent it**, see below.
> >
> > First, to produce $\mathbb{P}$, we use pre-trained WGAN-QC model. This is a technical thing needed only to sample from $\mathbb{P}$ and formally make it continuous. That is, we identify the **real data** distribution with $\mathbb{P}$ (*real good images*). At this step, one may employ any *other generative model* -- non-OT based GANs, normalizing flows, diffusion models, etc. -- it does not matter, as we just need some mechanism to sample from $\mathbb{P}$. Second, we use $\mathbb{P}$, 1-Lipschitz MinFunnel $u$ and construct $u$-ray monotone $\mathbb{Q}=T\sharp\mathbb{P}$ simulating a fake distribution (*fake bad images*), but in reality it is not produced by a WGAN.
> >
> > When testing OT solvers, we consider the OT gradient from bad images $\mathbb{Q}$ to good images $\mathbb{P}$ as it is intuitively more related to the WGAN setting. The benchmark pair can be tested in the other direction $(\mathbb{P}\rightarrow\mathbb{Q})$ as well.
> >
> >
> >
> >
> > **Concluding remarks**. Please respond to our post to let us know if the clarifications above suitably address your concerns about our work. We are happy to address any remaining points during the discussion phase.

---

> > ### Comment · Reviewer_PTCv · 2022-08-24
> > **Thanks for your clarifications**
> >
> > Dear authors,
> >
> > Thank you for your comprehensive response. I have read through all of them.
> >
> > About the question "If we carefully choose the MinFunnel function $u$, instead of randomly picking, will the performance be better?", I was just interested in approximating a specific $f^\*$ by $u$.
> >
> > For example, if model A is based on a specific 1-Lipschitz function $f_1^\*$, and our benchmark pairs are also approximating this specific $f_1^\*$ by $u$, then will model A beat other models due to it uses the same 1-Lipschitz function with the benchmark? If the answer is yes, are our benchmark pairs approximating one specific 1-Lipschitz function of one existing model by chance and leading to unfair evaluation?

---

> > > ### Author Response · Authors · 2022-08-29
> > > **Further clarifications**
> > >
> > > In general, we suppose the answer is negative. For example, consider two following functions $[0,1]\rightarrow \mathbb{R}$:
> > >
> > > $f^{\star}_{1}(x)=0$
> > >
> > > $u_{N}(x)=\min_{n=0,1,\dots,N} |x-\frac{n}{N}|$
> > >
> > > Both functions are $1$-Lipschitz (moreover, $f_{1}^{*}$ is 0-Lipschitz). It is clear that when $N\rightarrow \infty$, we have $|u-f_{1}^{\star}|\rightarrow 0$ in the supremum norm. However, $|\nabla f_{1}^{\star}(x)-\nabla u(x)|=1$ at every point $x\in [0,1]$ except for a negligible set. Thus, even though $u_{N}$ (for a large $N$) is a good approximation of $f_{1}^{\star}$, it does not provide a meaningful approximation of its gradient. In turn, $f_{1}^{\star}$ will always score $L^{2}$-UVP$=100$% (which is bad) on the benchmark constructed by $u$.
> > >
> > > P.S. This issue is called the **gradient deviation** and was mentioned in the related Wasserstein-2 Benchmark (see the end of section 2).
> > >
> > > [23] Korotin, A., Li, L., Genevay, A., Solomon, J. M., Filippov, A., & Burnaev, E. (2021). Do neural optimal transport solvers work? a continuous wasserstein-2 benchmark. Advances in Neural Information Processing Systems, 34, 14593-14605.

---

> > > > ### Comment · Reviewer_PTCv · 2022-09-02
> > > > **Thanks for the clarifications**
> > > >
> > > > Thanks for your further clarifications. I keep my decision and vote for acceptance.

---

### Official Review · Reviewer_Ceo9 · 2022-07-29
**Review of Kantorovich Strikes Back!**

**Rating:** 6
**Confidence:** 2

**Strengths:**

- **{S1}** While previous works use discrete distributions for benchmarking solvers, this work suggests continuous distributions, which is a novel aspect for benchmarking W_1.

**Weaknesses:**

- **{W1}** The benchmark contains only one image dataset with a single mode (faces). The addition of more image datasets, especially multi-modal ones (e.g. CIFAR-10), would improve the versatility of the benchmark and extend it to conditional models.

**Additional Feedback:**

See {W1}.

**Clarity:**

The writing itself is good, however, as someone coming from a different background it was hard to follow.

**Correctness:**

I am not an expert and have been working on the practical side before, so I cannot assess the correctness of this paper.

**Documentation:**

The code to use the benchmark is accessible via GitHub and documented there.

**Ethics:**

I can not see reasons for further ethical discussions.

**Relation To Prior Work:**

This benchmark suggests comparing continuous distributions, while former works compared discrete distributions.

**Summary And Contributions:**

This paper provides a benchmark to evaluate approximators for Wasserstein-1 distances as loss functions in the generative adversarial network setting.

---

> ### Author Response · Authors · 2022-08-18
> **Answer to Reviewer  Ceo9**
>
> Thank you for your valuable feedback. Please find above (in our reply to all the Reviewers) the answers to your comments common with other reviews. Please respond to our post to let us know if the clarifications above suitably address your concerns about our work. We are happy to address any remaining points during the discussion phase.

---

### Author Response · Authors · 2022-08-18
**General response**

Dear reviewers, we thank you for your insightful comments and interesting questions! We are glad that you positively highlight our theoretical insights (Reviewer PTCv), clear writing (Reviewer Givm, Reviewer Ceo9, Reviewer kVP9) and literature review (Reviewer CJQz), evaluation of many dual solvers (Reviewer 7yDG). Please find the answers to your shared questions below.

**(1) Considering real-world multi-modal image datasets, e.g., CIFAR-10. (Reviewers Ceo9,  CJQz)**

To address your comment, we trained a WGAN-QC [28] generator model for **multi-modal** CIFAR-10 dataset containing $32\times 32$ RGB images. Then, analogously to our CelebA benchmark pairs (Section 3.3), we employed our methodology (Section 3) to design **2 new CIFAR-10 images benchmark pairs** ($N=1,16$ funnels in dimension $D=3072$).

(1a) We added the description of the benchmarks pairs to the revised paper and will commit them the GitHub repo.

(1b) We evaluated the OT solvers on the benchmark and summarized the results in Tables 3, 4, 5 of appendix.

Our evaluation shows that the metrics of OT solvers on CIFAR-10 pairs are analogous to those on CelebA pairs.

**(2) The results of the benchmarks are only available in the Appendices (Reviewers CJQz, 7yDG)**

Unfortunately, there are too many evaluation results (Tables with metrics of solvers' performance) and moving them to the main text is not possible. Thus, we tried to summarize the results in the text discussion. In the revision, we moved the extended discussion of the solvers' performance in the images benchmark pairs from the Appendix to the main text. Additionally, in Section 4, we added a Table 1 which summarizes all the evaluation results.

**Paper revision**. We have uploaded the revised version of the paper. The newly added content is highlighted with the **blue** color. The parts which are moved from Appendix to the main text are in **purple**. The major changes are:

**[Ceo9,  CJQz]** We constructed new benchmark pairs based on CIFAR-10 dataset and used them to evaluate OT solvers (Section 3.3, Figure 6 in the main text; Tables 2, 3, 4 in Appendix D).

**[CJQz, 7yDG]** We extended the discussion of the evaluation results on the images pairs (lines 299-313 of Section 4).

**[Givm, kVP9]** We extended the discussion of limitations of the evaluation (Section 4).


We are currently working on adding the new changes (CIFAR-10 benchmark pairs) to the GitHub repo.

**Concluding remarks.** Please respond to our post to let us know if the clarifications above suitably address your concerns about our work. We are happy to address any remaining points during the discussion phase.

---

> ### Author Response · Authors · 2022-08-29
> **Repository update (CIFAR-10 benchmark pairs)**
>
> Dear reviewers, please consider the updated [repository](https://github.com/justkolesov/Wasserstein1Benchmark). Now it contains the newly constructed CIFAR-10 images benchmark pairs (see our previous post).

---

### Meta-Review · Area_Chair_5t5X · 2022-09-07

**Recommendation:** Accept
**Confidence:** 3

**Metareview:**

This paper proposes a new benchmark to evaluate the solution of optimal transport problems. The reviewers concur that the benchmark is well-executed and novel. Some are concerned that a better benchmark for OT problems will not drive progress, as the successes of Wasserstein GANs occur despite their failure to solve OT. However, it seems like a useful intermediate check to deepen understanding of why Wasserstein GANs (and models to come!) work by (at least) eliminating non-explanations.

---

### Decision · Program_Chairs · 2022-09-16

Accept